# A Survey on Self-Play Methods in Reinforcement Learning

## Abstract

Self-play, a learning paradigm where agents iteratively refine their policies by interacting with historical or concurrent versions of themselves or other evolving agents, has shown remarkable success in solving complex non-cooperative multi-agent tasks. Despite its growing prominence in multi-agent reinforcement learning (MARL), such as Go, poker, and video games, a comprehensive and structured understanding of self-play in non-cooperative games remains lacking. This survey fills this gap by offering a comprehensive roadmap to the diverse landscape of self-play methods in non-cooperative games. We begin by introducing the necessary preliminaries, including the MARL framework and basic game theory concepts. Then, it provides a unified framework and classifies existing self-play algorithms within this framework. Moreover, the paper bridges the gap between the algorithms and their practical implications by illustrating the role of self-play in different non-cooperative scenarios. Finally, the survey highlights open challenges and future research directions in self-play.

## 1 Introduction

Reinforcement learning (RL) represents a significant paradigm (Sutton & Barto, 2018) within machine learning (ML), focused on optimizing decision processes through interaction with an environment. In RL, the environment is modeled as a Markov decision process (MDP), where agents observe states, take actions, receive rewards, and cause transitions to new states. The primary goal of RL algorithms is to derive the optimal policy that yields the maximum expected accumulated reward over time. Deep RL extends traditional RL by employing deep neural networks as function approximators to handle high-dimensional state spaces, contributing to breakthroughs in various complex tasks (Mnih et al., 2013).

Transitioning from single-agent to multi-agent reinforcement learning (MARL) introduces complex dynamics (Rashid et al., 2018; Mahajan et al., 2019; Yu et al., 2022). In MARL, the interdependence of agents' actions presents significant challenges, as the environment appears non-stationary to each agent. In the context of MARL, environments can generally be categorized into cooperative and non-cooperative settings. In cooperative scenarios, all agents share a common objective and are incentivized to work together, often leading to formulations that maximize a joint reward. In contrast, non-cooperative settings involve agents with individual or even conflicting goals, where the success of one agent may come at the expense of others. In MARL, non-cooperative tasks are generally harder than cooperative ones because the absence of a shared reward prevents the effective use of centralized training, a key technique for mitigating non-stationarity in cooperative settings (Foerster et al., 2016; 2018; Lowe et al., 2017; Yu et al., 2022), thereby further aggravating non-stationarity.

With the help of game theory, a mathematical framework that models the interactions between multiple decision-makers, self-play emerges as an elegant solution to some inherent challenges in MARL. Vanilla self-play refers to a setting in which an agent continuously trains by competing against the most recent version of itself, as famously demonstrated for the symmetric zero-sum game of checkers (Samuel, 1959). Self-play, however, encompasses far more than this canonical example. More broadly, self-play refers to a class of learning methods where at least one agent iteratively improves its policy by interacting with a distribution of evolving opponents, which may include historical or concurrent versions of itself or other agents. It is worth noting that although some MARL studies have applied the concept of self-play in cooperative tasks such as Hanabi (Cui et al., 2021) and Overcooked (Strouse et al., 2021), our survey is conducted within

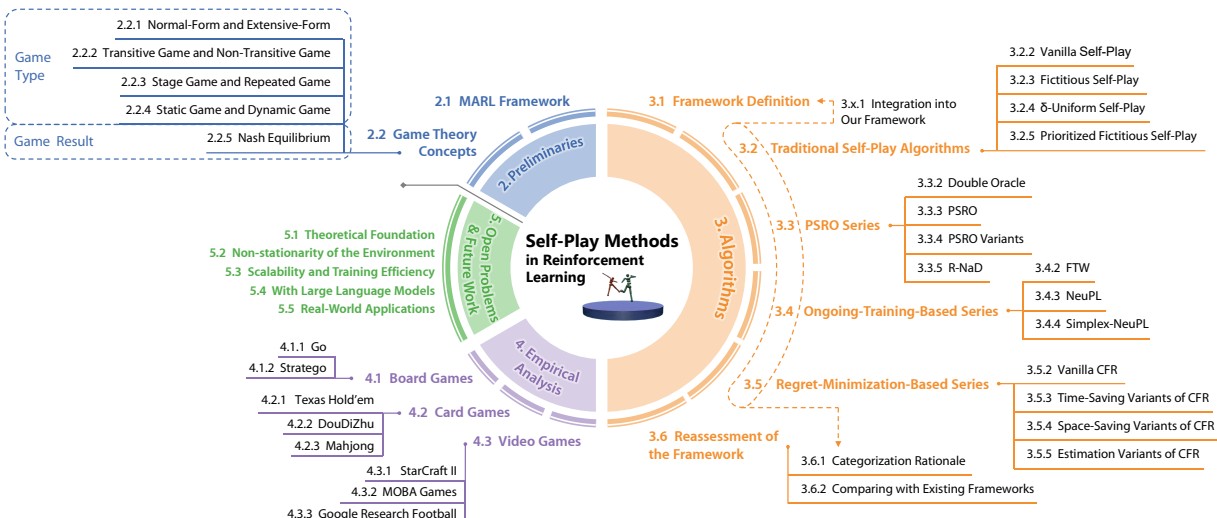

Figure 1: Overview of our survey.

non-cooperative settings. Self-play has demonstrated remarkable success in non-cooperative domains such as Go (Silver et al., 2016; 2017; 2018; Schrittwieser et al., 2020), chess (Silver et al., 2018; Schrittwieser et al., 2020), poker (Moravčík et al., 2017; Heinrich et al., 2015), and complex video games (Berner et al., 2019; Vinyals et al., 2019). In these scenarios, it has developed strategies that surpass human expertise. Although the application of self-play is extensive and promising, it is accompanied by limitations, such as the potential convergence to suboptimal strategies and significant computational requirements (Silver et al., 2016; 2018).

Although some research (Wellman et al., 2025) adopts a different perspective through empirical game-theoretic analysis (EGTA), relatively few comprehensive surveys focus on self-play. Among them, a study (Hernandez et al., 2021) develops an algorithmic framework for self-play, but it does not incorporate the Policy-Space Response Oracle (PSRO) (Lanctot et al., 2017) series of algorithms, while a separate study (Bighashdel et al., 2024) focuses exclusively on PSRO, without considering other self-play algorithms. Another study (DiGiovanni & Zell, 2021) examines the theoretical safety of self-play. While these studies are valuable, they do not provide a comprehensive perspective that fully captures the breadth and depth of self-play. To address this gap, this survey aims to offer a systematic and holistic roadmap to the diverse landscape of self-play algorithms in MARL within non-cooperative settings.

The survey is organized as follows. Sec. 2 introduces the background of self-play, including the MARL framework and game theory concepts. Sec. 3 proposes a unified framework and then categorizes existing self-play algorithms into four categories based on this framework. In Sec. 4, a comprehensive analysis illustrates how self-play is applied in various scenarios. Sec. 5 describes open problems in self-play and explores future research directions. Finally, Sec. 6 concludes the survey on self-play. A more detailed overview of our survey is depicted in Fig. 1, which further illustrates the relationships among the subsections.

## 2 Preliminaries

This section first introduces the framework of MARL, in which agents interact with an environment to learn policies that maximize cumulative rewards. We then present fundamental concepts from non-cooperative game theory, which provides a formal framework for analyzing strategic interactions among rational decision-makers who pursue individual objectives.

## 2.1 MARL Framework

In RL, the environment is modeled as an MDP, where the Markovian assumption states that the environment's evolution is fully determined by its current state, eliminating the need to consider past states. An agent interacts with the environment by taking actions, which result in different states and associated rewards. MDPs can be extended to multi-agent settings, known as Markov games (MGs) (Littman, 1994) or stochastic games (Shapley, 1953). We focus on the most general form: partially observable Markov games (POMGs), where multiple agents interact with the environment, and each agent receives only individual observations instead of the full state, meaning they have limited access to the environment's state.

A POMG $\mathcal{G}$ is defined by $\mathcal{G} = (\mathcal{N}, \mathcal{S}, \mathcal{A}, \mathcal{O}, \mathcal{P}, \mathcal{R}, \gamma, \rho)$. $\mathcal{N} = \{1, \cdots, n\}$ denotes n agents. $\mathcal{S}$ is the state space. $\mathcal{A} = \prod_{i=1}^{n} \mathcal{A}_i$ is the product of the action space of each agent. Similarly, $\mathcal{O} = \prod_{i=1}^{n} \mathcal{O}_i$ is the product of the observation space of each agent. $\mathcal{P} : \mathcal{S} \times \mathcal{A} \times \mathcal{S} \rightarrow [0, 1]$ denotes the transition probability from one state to another given the actions of each agent. $\mathcal{R} = \{\mathcal{R}_1, \cdots, \mathcal{R}_n\}$, where $\mathcal{R}_i : \mathcal{S} \times \mathcal{A} \rightarrow \mathbb{R}$ denotes the reward function of agent $i$. $\gamma \in [0, 1]$ is the discount factor. $\rho : \mathcal{S} \rightarrow [0, 1]$ describes initial state distribution.

More concretely, in RL, agents interact with the environment through the following procedure: At each discrete time step $t$, each agent $i$ receives an observation $o_{i,t}$ from the environment and selects an action $a_{i,t}$ based on a stochastic policy $\pi_i : \mathcal{O}_i \times \mathcal{A}_i \rightarrow [0, 1]$. After receiving the joint actions $\mathbf{a}_t = (a_{1,t}, \cdots, a_{n,t})$, the environment transitions from the current state $s_t$ to a subsequent state $s_{t+1}$ according to the transition function $\mathcal{P}$ and sends a reward $r_{i,t+1}$ to every agent $i$. The ultimate goal of agent $i$ is to learn a policy $\pi_i$ that maximize the expected discounted cumulative rewards: $\mathbb{E}_{(\pi_i, \pi_{-i})}[\sum_{t=0}^{\infty} \gamma^t r_{i,t+1}]$, where $\pi_{-i}$ denotes the other agents' policies.

## 2.2 Game Theory Concepts

### 2.2.1 Normal-Form and Extensive-Form

The normal form and extensive form are two distinct representations of games in game theory. If a game $\mathcal{G}$ is represented in the **normal-form**, it can be expressed by $\mathcal{G} = (\mathcal{N}, \mathbf{\Pi}, \mathbf{u})$. $\mathcal{N} = \{1, 2, \cdots, n\}$ denotes the players. The set $\mathbf{\Pi} = \Pi_1 \times \cdots \times \Pi_n$ represents the pure strategy space for all players. A **pure strategy** specifies a deterministic action choice for each point in the game. A vector $\boldsymbol{\pi} = (\pi_1, \ldots, \pi_n) \in \mathbf{\Pi}$ is called a **strategy profile**. A **mixed strategy** $\boldsymbol{\sigma}$ assigns a probability distribution over the set of pure strategies. For player $i$, a mixed strategy is represented by $\sigma_i \in \Delta(\Pi_i)$, where $\Delta$ denotes the probability simplex. $\mathbf{u} = (u_1, \cdots, u_n)$, where $u_i : \mathbf{\Pi} \rightarrow \mathbb{R}$, is a **utility function** that assigns the expected **payoff** to each player $i$. If $\exists C \in \mathbb{R}, \forall \boldsymbol{\pi} \in \mathbf{\Pi}, \sum_i u_i(\boldsymbol{\pi}) = C$, the game is a **constant-sum** game. Specially, if $C = 0$, it is a **zero-sum** games, in which every gain by one player is exactly offset by losses to the others. If no such constant $C$ exists, it is a **general-sum** game. Moreover, if $\Pi_1 = \cdots = \Pi_n$ and for any permutation $\tau$ and any strategy profile $(\pi_1, \ldots, \pi_n)$, $u_i(\pi_1, \cdots, \pi_n) = u_{\tau(i)}(\pi_{\tau(1)}, \cdots, \pi_{\tau(n)})$, the game is a **symmetric game**. If a finite set of players is involved and each player has a finite set of strategies, a normal-form game can be depicted in a tensor $T \in \mathbb{R}^{|\Pi_1| \times \cdots \times |\Pi_n| \times n}$, where $|\Pi_i|$ is the size of player $i$'s strategy space and the final dimension indexes each player's payoff.

Specifically, in *two-player zero-sum symmetric normal-form games*, both players share the same pure strategy space, denoted by $\Pi$, such that $\Pi = \Pi_1 = \Pi_2$. Since the utility function satisfies $u_1(\pi_i, \pi_j) = -u_2(\pi_i, \pi_j)$, we can simplify the utility to a single function $u$, where for $\pi_i, \pi_j \in \Pi$, if $\pi_i$ beats $\pi_j$, then $u(\pi_i, \pi_j) = -u(\pi_j, \pi_i) > 0$. This game can be directly depicted in an **evaluation matrix** $M_\Pi \in \mathbb{R}^{|\Pi| \times |\Pi|}$. It captures the game outcomes by detailing the results of different strategies when they are played against each other: $M_\Pi = \{u(\pi_i, \pi_j) : \pi_i, \pi_j \in \Pi \times \Pi\}$.

If a game is represented in the **extensive-form**, it is expressed sequentially, illustrating the sequence of moves, choices made by the players, and the information available to each player during decision-making. Typically, a game in the extensive-form is represented by a game tree. This tree demonstrates the sequential and potentially conditional nature of decisions. A game $\mathcal{G}$ represented in the extensive-form can be expressed by $\mathcal{G} = (\mathcal{N} \cup \{c\}, H, Z, P, \mathcal{I}, A, \mathbf{u})$. $\mathcal{N} = \{1, 2, \cdots, n\}$ denotes a set of players. $c$ is the **chance** node and can be regarded as a special player. $H$ represents a set of possible **histories** and $Z \subseteq H$ is a set of **terminal histories**. Order of moves is represented by a function $P(h) \in \mathcal{N} \cup \{c\}$ to indicate which player is to move,

where $h \in H$. $\mathcal{I}$ denotes **information set partitions** and $\mathcal{I}_i$ denotes the information set partitions for player $i$. If player $i$ reaches a history $h \in I_i$, where $I_i \in \mathcal{I}_i$ is a specific **information set** for player $i$, all decision nodes in $I_i$ are observationally equivalent to player $i$, and he cannot tell which specific node within $I_i$ has been reached. A game has **perfect information** precisely when every information set of every player is a singleton. Formally, $\forall i, \ \forall I_i \in \mathcal{I}_i, \ |I_i| = 1$. In such games, whenever a player is to move, he knows exactly which decision node has been reached and, equivalently, the full sequence of past actions and observed states. Otherwise the game is considered to have **imperfect information**. Action space is represented by $A(h)$ for a non-terminal history $h \in H$. For all non-terminal histories $h$ within an information set $I_i$, the available actions are the same; otherwise, they are distinguishable. Therefore, we use $A(I_i)$ to represent the available actions for the information set $I_i$. Utility functions is denoted by $\mathbf{u} = (u_1, \cdots, u_n)$, where $u_i : Z \to \mathbb{R}$. Together, these components define the structure and dynamics of an extensive-form game. Moreover, a **behavior strategy profile** can be expressed by $\boldsymbol{\beta} = (\beta_1, \cdots, \beta_n)$, where $\beta_i$ maps each $I_i \in \mathcal{I}_i$ to a probability distribution over $A(I_i)$. A game satisfies **perfect recall** if each player $i$ remembers the entire sequence of his past actions and all of the information he had at each of his previous decision points. In any finite extensive-form game with perfect recall, for every mixed strategy $\boldsymbol{\sigma}$, there exists an equivalent behavior strategy $\boldsymbol{\beta}$, and vice versa, such that both strategies induce the same distribution over terminal histories (i.e., the same outcomes in the game tree). A **subgame** of an extensive-form game is any subset of the game tree that (i) has exactly one initial decision node, (ii) whenever it contains a node it also contains all of its descendants, and (iii) whenever it contains a node it contains every other node in that node's information set.

Beyond normal-form and extensive-form games, the analysis of complex games often involves a higher-level abstraction: the **meta-game**. A meta-game provides an abstraction over the original game by modeling interactions between entire policies rather than individual actions. While commonly instantiated as a normal-form game, a meta-game can, in general, take any game-theoretic form (e.g., extensive-form) as long as it captures strategic interactions over a policy population. In this setting, players choose policies, and **meta-strategies** represent mixed strategies over these policies.

### 2.2.2 Transitive Game and Non-Transitive Game

To provide a clearer introduction to transitive and non-transitive games, in this section, we confine our analysis to two-player zero-sum symmetric games. In a **transitive game**, the strategies or outcomes follow a transitive relationship. Formally, $\forall \pi_i, \pi_j, \pi_k \in \Pi$, if $u(\pi_i, \pi_j) > 0$ and $u(\pi_j, \pi_k) > 0$, then it must follow that $u(\pi_i, \pi_k) > 0$. This transitive property simplifies the strategic landscape, allowing for an ordinal ranking of strategies. Conversely, in a **non-transitive game**, $\exists \pi_i, \pi_j, \pi_k \in \Pi$ such that $u(\pi_i, \pi_j) > 0$ and $u(\pi_j, \pi_k) > 0$, but $u(\pi_i, \pi_k) \leq 0$. This introduces a cyclic relationship among strategies, thereby complicating the game. The complexity often results in a mixed-strategy equilibrium, where players randomize their choices among multiple strategies to maximize their expected payoff. A classical example of a non-transitive game is Rock-Paper-Scissors, in which no single strategy uniformly dominates all others.

### 2.2.3 Stage Game and Repeated Game

A **stage game** (or **one-shot game**) is a game that is played only once, namely a one-shot interaction between players. A famous example of a stage game is the Prisoner's Dilemma. A **repeated game** is derived from a stage game played multiple times. Formally, a repeated game based on a stage game $\mathcal{G}$ is defined by playing $\mathcal{G}$ for $T$ periods, where $T$ can be finite or infinite. The strategies in a repeated game are history-contingent, meaning they can depend on the entire sequence of past plays. An example of a repeated game is Texas Hold'em, where players engage in multiple rounds, and each player's strategy may evolve based on previous rounds, betting patterns, and the history of interactions among players.

### 2.2.4 Nash Equilibrium

For simplicity, $\pi_i$ denotes the strategy of player $i$, and $\pi_{-i}$ denotes the strategies of all players other than player $i$. Given $\pi_{-i}$, player $i$'s **best response (BR)** is the strategy that maximizes player $i$'s payoff:

$$BR_i(\pi_{-i}) = \arg\max_{\pi_i} u_i(\pi_i, \pi_{-i}). \tag{1}$$

A strategy $\pi_i^*$ is an $\boldsymbol{\epsilon}$-**BR** to strategies $\pi_{-i}$ if:

$$u_i(\pi_i^*, \pi_{-i}) \geq u_i(BR_i(\pi_{-i}), \pi_{-i}) - \epsilon, \tag{2}$$

where $\epsilon$ is a pre-specified threshold.

A strategy profile $(\pi_1^*, \pi_2^*, ..., \pi_n^*)$ is a **Nash equilibrium (NE)** if, for each player $i$:

$$u_i(\pi_i^*, \pi_{-i}^*) \geq u_i(\pi_i, \pi_{-i}^*), \forall \pi_i, \tag{3}$$

meaning that no player can benefit by changing their strategy unilaterally, given the strategies of others. In other words, an NE is a situation where each player's strategy is a BR to the others' strategies.

A strategy profile $(\pi_1^*, \pi_2^*, ..., \pi_n^*)$ is an $\boldsymbol{\epsilon}$-**NE** if, for each player $i$:

$$u_i(\pi_i^*, \pi_{-i}^*) \geq u_i(\pi_i, \pi_{-i}^*) - \epsilon, \forall \pi_i, \tag{4}$$

meaning that no player can increase their payoff by more than $\epsilon$ by unilaterally changing their strategy.

However, computing NE is generally intractable in complex games, so some researchers utilize $\alpha$-Rank (Omidshafiei et al., 2019) and Correlated Equilibrium (CE) (Aumann, 1974) as alternatives. Some studies also resort to Replicator Dynamics (Taylor & Jonker, 1978) to analyze strategy evolution.

### 2.2.5 Static Game and Dynamic Game

The market entry game is a classic example in game theory. There are two players: Firm 1 (the entrant), which is considering whether to enter a market, and Firm 2 (the incumbent), which currently dominates that market. The game outcomes are as follows:

- If the entrant stays out of the market, the entrant earns a payoff of 4, and the incumbent earns 10.
- If the entrant enters and the incumbent chooses to attack, both firms receive a payoff of 3.
- If the entrant enters and the incumbent chooses not to attack, both firms receive a payoff of 6.

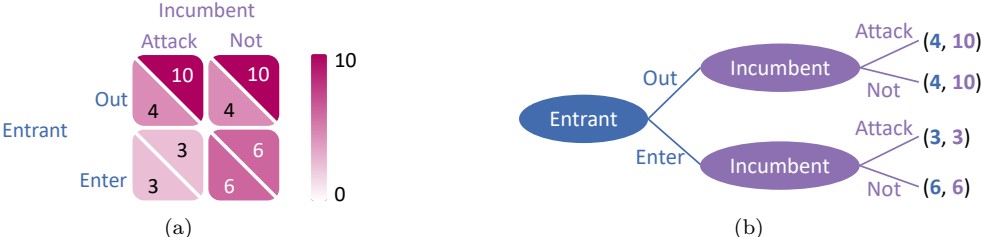

(a)             (b)

Figure 2: The example of the market entry game. (a) Matrix representation of *simultaneous* market entry game in normal-form. (b) Game tree representation of *sequential* market entry game in extensive-form.

In the *simultaneous* version of the market entry game, both players choose their strategies at the same time. The entrant decides whether to enter, and the incumbent simultaneously decides whether to attack, without knowing the entrant's decision. Games of this kind are referred to as **static games**. Although a static game can be represented in either normal- or extensive-form, it is conventionally shown in the normal form because it succinctly captures the strategic structure of simultaneous moves (as shown in Fig. 2a). In the *sequential* version, the entrant moves first, and the incumbent responds after observing the entrant's decision. Games of this nature are called **dynamic games**. Although any dynamic game admits a normal-form representation, that representation scales exponentially with the number of information sets. Consequently, dynamic games are typically illustrated in extensive form, whose game tree compactly encodes the order of moves and the information available to each player (as depicted in Fig. 2b).

The market entry game demonstrates how the structure of a game influences equilibrium outcomes. In the static version, non-credible threats—such as the incumbent's threat to attack even when it is not in its best interest—may support a Nash equilibrium where the entrant stays out. In contrast, the dynamic version allows for refinement through **subgame perfect Nash equilibrium (SPNE)**, which requires that players' strategies constitute a Nash equilibrium in every subgame. Under this refinement, the threat to attack is ruled out as non-credible, and the predicted outcome is that the entrant will enter and the incumbent will accommodate, since attacking would harm the incumbent itself.

## 3 Algorithms

Based on existing self-play work (Hernandez et al., 2021; Lanctot et al., 2017; Liu et al., 2022c; Garnelo et al., 2021), we propose a self-play framework (Algo. 1) that boasts enhanced expressivity and superior generalization capabilities. For simplicity, our framework is illustrated for *symmetric games*. All players share a policy population with a fixed maximum size. In each iteration, a newly initialized policy is trained, while opponent policies are sampled from the population. After training, the new policy is added to the population, possibly replacing an existing policy. The updated policy population is then evaluated, and then the opponent sampling strategy is recalculated for the next iteration.

This section is organized as follows: In Sec.3.1, we provide a formalized description of our framework. We then categorize self-play algorithms into four primary groups: traditional self-play algorithms (Sec. 3.2), the PSRO series (Sec. 3.3), the ongoing-training-based series (Sec. 3.4), and the regret-minimization-based series (Sec. 3.5). We analyze how these four categories align with our framework and introduce corresponding algorithms for each category. To make it more straightforward, we highlight the representative classic algorithms from these and present them in Table 1 for comparison within our framework. Sec. 3.6 compares the four categories, explains our classification rationale, contrasts the framework with prior ones to demonstrate its greater generality, and highlights the design insights it offers for future self-play research.

### 3.1 Framework Definition

---
**Algorithm 1** A unified framework of self-play.

---
1: Initialize $\Pi, \Sigma$
2: **for** $e \in [[E]]$ **do**
3:     **for** $\sigma_{[k]} \in [[K]]$ **do**
4:         Initialize $\pi_{[k]}^h$
5:         $\pi_{[k]}^h \leftarrow ORACLE(\pi_{[k]}^h, \sigma_{[k]}, \Pi)$            ▷ Compute the oracle.
6:         $\mathcal{P} \leftarrow EVAL(\Pi)$            ▷ Get policies' performance (Optional).
7:         $\Sigma \leftarrow MSS(\mathcal{P})$            ▷ Update the interaction matrix (Optional).
8:     **end for**
9: **end for**
10: **return** $\Pi, \Sigma$

---

In Algo. 1, we define a unified self-play framework based on Liu et al. (2022c); Lanctot et al. (2017); Garnelo et al. (2021); Hernandez et al. (2021). We combine visual illustrations (Fig. 3) with detailed textual descriptions to explain the key processes of our framework. Next, we provide an in-depth explanation of these processes:

- $\Pi := \{\pi_{[k]}(\cdot|h(k))\}_{k=1}^K$: Each policy $\pi_{[k]}$ in the **policy population** $\Pi$ is conditioned on a **policy condition function** $h(k)$, which provides supplementary information for certain algorithms. We denote the policy $\pi_{[k]}(\cdot|h(k))$ as $\pi_{[k]}^h$. Note that $\pi_{[k]}$ refers to the $k$th policy in the population, and $K$ denotes the **policy population size**. $\Pi$ can be initialized (Line 1 in Algo. 1) in two ways: lazy initialization and immediate initialization. In **lazy initialization**, $\Pi$ starts with $K$ placeholder policies, with the actual policies being initialized during the training iterations (Line 4 in Algo. 1).

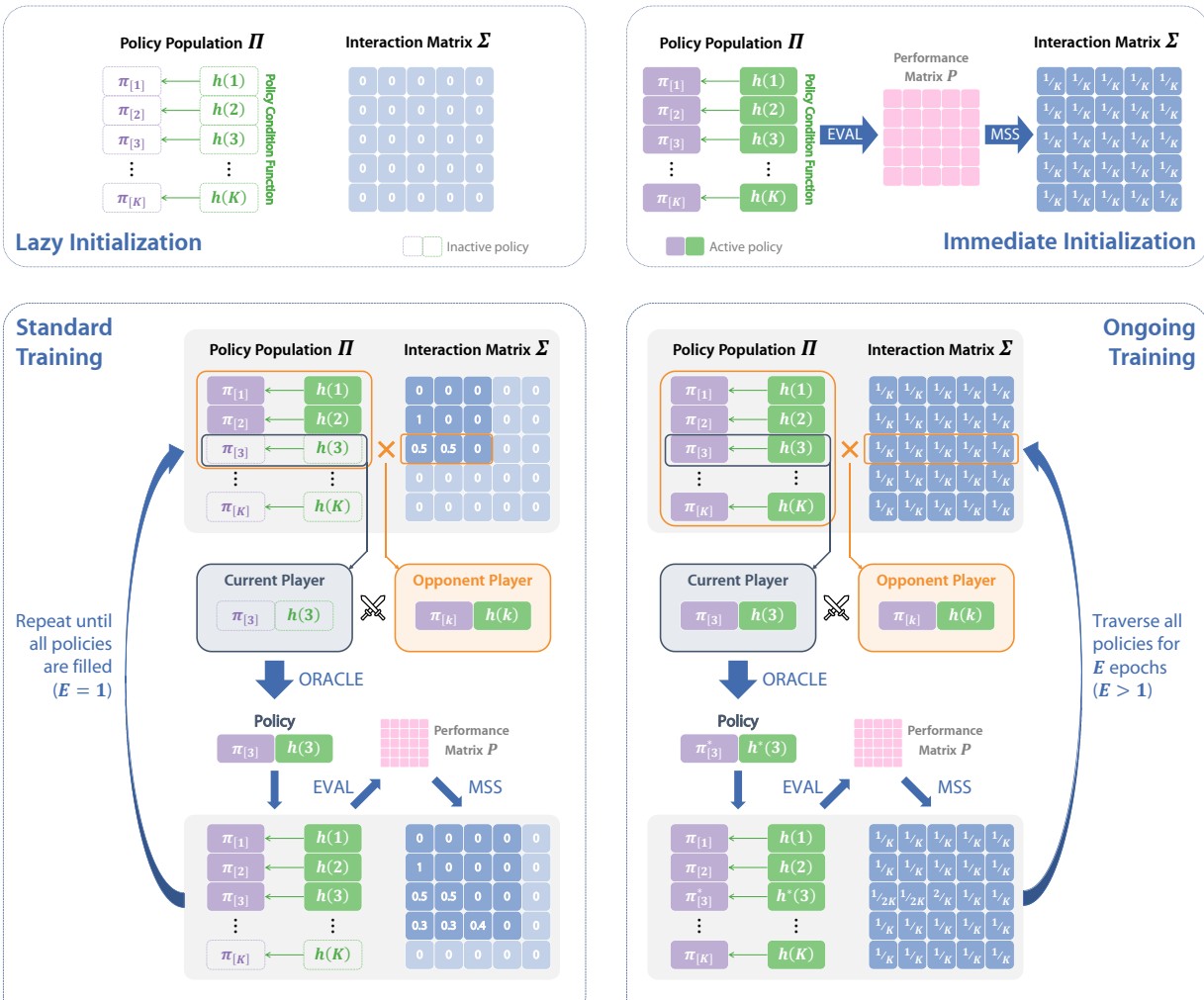

Figure 3: Illustration of our framework. The top row shows two different initialization methods for the policy population $\Pi$ and the interaction matrix $\Sigma$: lazy initialization and immediate initialization (corresponding to Line 1 in Algo. 1). The bottom row illustrates two training paradigms: standard training and ongoing training (corresponding to Lines 2~9 in Algo. 1). We categorize self-play algorithms into four types: traditional self-play, the PSRO series, the ongoing-training-based series, and the regret-minimization-based series. Among these, traditional self-play, the PSRO series, and the regret-minimization-based series utilize lazy initialization and standard training, whereas the ongoing-training-based series employs immediate initialization and ongoing training. For a detailed description and analysis, please refer to Sec.3 and Table1.

In **immediate initialization**, $\Pi$ is initialized upfront with $K$ real policies, which may be randomly generated or pre-trained models. A policy $\pi_{[k]}^h$ is considered **inactive** if it is a placeholder; otherwise, it is considered **active**.

- $\Sigma := \{\sigma_{[k]}\}_{k=1}^{K+1} \in \mathbb{R}^{K \times C_1}$: The **interaction matrix** $\Sigma$ consists of rows $\sigma_{[k]}$. Each $\sigma_{[k]} \in \mathbb{R}^{C_1}$ represents the probability distribution over opponent policies in the policy population $\Pi$, where $C_1 = K^{n-1}$ and $n$ denotes the number of players. Specially, in a two-player game, if $C_1 = K$ and $\Sigma_{mn}$ represents the probability that policy $m$ is optimized against policy $n$, $\Sigma$ can be depicted in **directed interaction graphs** (Fig. 4). $\Sigma$ will be initialized with all zeros (Line 1 in Algo. 1), providing a starting point before the meta-strategy solver updates it using the first payoff information.

- For epoch $e \in [[E]]$ (Line 2 in Algo. 1): $E$ denotes the total number of epochs for the entire policy population. For example, if the algorithm only introduces new policies into the population without updating existing ones, then $E = 1$. This implies that only the *inactive* policy is likely to be chosen for training in each iteration and turned into an *active* one, while *active* policies remain unchanged. Conversely, if *active* policies are updated multiple times throughout the algorithm, then $E > 1$. Indeed, $E$ accurately reflects the number of updates performed.

- Initialize $\pi^h_{[k]}$ (Line 4 in Algo. 1): The initialization of $\pi^h_{[k]}$ can vary depending on the algorithm being used. It can be initialized randomly or by leveraging pre-trained models (Silver et al., 2016; Vinyals et al., 2019) or through a recently updated policy.

- ORACLE($\pi_{[k]}, \sigma_{[k]}, \Pi$) (Line 5 in Algo. 1): The ORACLE is an abstract computational entity that returns a new policy adhering to specific criteria. Here, we divide ORACLE into three types. (1) One type is the BR oracle, which is designed to identify the optimal counter-strategies against an opponent's strategy, including finding NE (McMahan et al., 2003). However, it often requires considerable computational effort. (2) To alleviate the computational demands, the approximate best response (ABR) oracle is introduced, using techniques such as RL (Algo. 2), evolution-theory-based methods (Algo. 3) or regret minimization methods (Algo. 4). (3) Other specially crafted ORACLEs are either tailored to specific meta-strategy solvers (MSSs) (Muller et al., 2020; Marris et al., 2021) or introduced to enhance diversity (Balduzzi et al., 2019; Perez-Nieves et al., 2021; Liu et al., 2021). Some details of Algo. 2, 3 and 4 need to be mentioned. Algo. 2 is a RL method that can be widely used across different categories. Moreover, on-policy RL algorithms do not require a sample buffer to collect trajectories. In contrast, only off-policy methods rely on a sample buffer to store and reuse past experiences (Line 11 and 15 in Algo. 2). Algo. 3 is specifically utilized by Regularized Nash Dynamics (R-NaD) (Perolat et al., 2021) and Algo. 4 is specifically utilized by the regret-minimization-based series. They will be introduced in detail in Sec. 3.3.5 and Sec. 3.5 respectively.

- EVAL($\Pi$) (Line 6 in Algo. 1): Evaluating the policy population $\Pi$. There are multiple evaluation metrics available to assess the performance of each policy. The performance matrix is represented as $\mathcal{P} := \{p_k\}_{k=1}^K \in \mathbb{R}^{K \times C_2}$. $p_k$ is the performance of policy k, and the dimension $C_2$ depends on the evaluation metric used. For instance, $p_k$ can be depicted as the relative skill like Elo ratings ($C_2 = 1$) or can be depicted as the payoff tensor ($C_2 = K^{n-1}$, where n is the number of players). Specially, in two-player symmetric zero-sum games, the expected payoffs can serve as the evaluation metric. In such cases, $\mathcal{P}$ is a square matrix ($C_2 = K$).

- $MSS : \mathbb{R}^{K \times C_2} \rightarrow \mathbb{R}^{K \times C_1}$ (Line 7 in Algo. 1): A **meta-strategy solver (MSS)** takes the **performance matrix** $\mathcal{P}$ as its input and produces a new interaction matrix $\Sigma$ as its output. If the MSS produces a constant matrix irrespective of the input, the evaluation step (Line 6 in Algo. 1) can be skipped to reduce computations. In this case, the MSS step also can be skipped. Additionally, if it is the final iteration, the MSS step should also be skipped, as the resulting meta-strategy will not be used. Moreover, if $\Pi$ uses immediate initialization, $\Sigma$ will be initialized (Line 1 in Algo. 1) by first evaluating the performance and then applying the MSS.

## 3.2 Traditional Self-Play Algorithms

Traditional self-play algorithms involve agents improving their strategies by repeatedly playing against themselves, allowing them to explore various strategies and enhance their decision-making abilities without external input. The simplest form involves agents training against their most recent version to identify weaknesses. Other approaches involve training against a set of strategies from different iterations, enabling agents to develop robust and adaptive strategies. This section will explain how traditional self-play algorithms fit into our framework and introduce representative traditional self-play methods, ranging from simpler forms to more complex ones.

---

**Algorithm 2** Compute the oracle in RL.

---

**Require:** $\pi_{[k]}^h$                   ▷ Policy $k$ is being trained.
**Require:** $\sigma_{[k]}$              ▷ Opponent sampling strategy of policy $k$.
**Require:** $\Pi$                    ▷ Policy population.
**Require:** $B$                  ▷ Sample Buffer (Optional).
 1: **while** $\pi_{[k]}^h$ is not valid **do**
 2:   **for** trajectory $\tau \in [[\tau_{max}]]$ **do**
 3:    sample $\boldsymbol{\pi}_{opp}^h \sim P(\sigma_{[k]})$            ▷ Policies of opponents.
 4:    $\boldsymbol{\pi} = (\pi_{[k]}^h, \boldsymbol{\pi}_{opp}^h)$
 5:    $s_0, \boldsymbol{o_0} \sim \rho$                 ▷ Initial state.
 6:    **for** $t \in [[t_{max}]]$ **do**
 7:     $\boldsymbol{a_t} \sim \boldsymbol{\pi}(\boldsymbol{o_t})$
 8:     $s_{t+1}, \boldsymbol{o_{t+1}} \sim P(s_t, \boldsymbol{a_t})$
 9:     $\boldsymbol{r_t} \leftarrow \boldsymbol{R}(s_t, \boldsymbol{a_t})$
10:    **end for**
11:    $B \leftarrow B \cup \{\tau\}$          ▷ Store trajectory in buffer (Optional).
12:    $\pi_{[k]}^h \leftarrow \text{update}(\pi_{[k]}^h)$           ▷ Using RL algorithms
13:   **end for**
14: **end while**
15: **return** $\pi_{[k]}^h, B$            ▷ Returning $B$ is optional.

---

**Algorithm 3** Compute the oracle in evolution theory.

---

**Require:** $\pi_{[k]}^h$                   ▷ Policy $k$ is being trained.
**Require:** $\sigma_{[k]}$              ▷ Opponent sampling strategy of policy $k$.
**Require:** $\Pi$                    ▷ Policy population.
 1: sample $\boldsymbol{\pi}_{opp}^h \sim P(\sigma_{[k]})$           ▷ Policies of opponents.
 2: $\boldsymbol{\pi}_{\text{reg}} = (\pi_{[k]}^h, \boldsymbol{\pi}_{opp}^h)$           ▷ Regularization policies.
 3: Transform the reward by adding a policy-dependent reward according to $\boldsymbol{\pi}_{\text{reg}}$ (Perolat et al., 2021).
 4: $\pi_{[k]}^h \leftarrow$ Use replicator dynamics to play the reward-transformed game until convergence.
 5: **return** $\pi_{[k]}^h$

---

**Algorithm 4** Compute the oracle in regret matching.

---

**Require:** $\pi_{[k]}^h$                   ▷ Policy $k$ is being trained.
**Require:** $\sigma_{[k]}$              ▷ Opponent sampling strategy of policy $k$.
**Require:** $\Pi$                    ▷ Policy population.
 1: **for** each player i **do**
 2:   sample $\boldsymbol{\pi}_{opp}^h \sim P(\sigma_{[k]})$           ▷ Policies of opponents.
 3:   $\boldsymbol{\pi} = (\pi_{[k]}^h(i), \boldsymbol{\pi}_{opp}^h(-i))$
 4:   Use regret matching to play the game and update $h(k)$ with new regret minimization information.
 5: **end for**
 6: **return** $\pi_{[k]}^h$

---

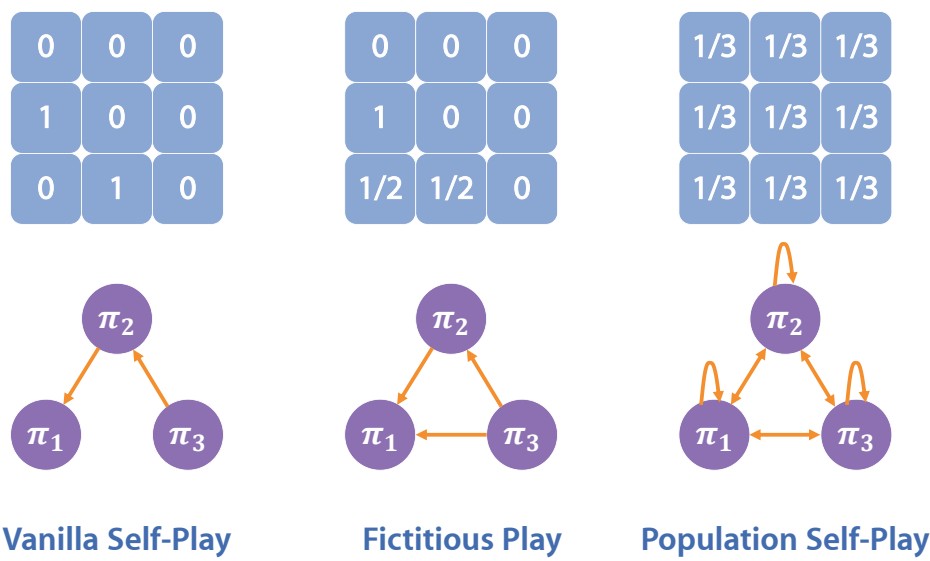

Figure 4: Interaction matrix $\Sigma$ examples. When $C_1 = K$ and the opponent sampling strategy $\Sigma_{mn}$ represents the probability that policy $m$ is optimized against policy $n$, the interaction matrix $\Sigma$ can be depicted in directed interaction graphs. Here, we consider three representative self-play algorithms. In the **Top** section, we define the interaction matrix $\Sigma \in \mathbb{R}^{3 \times 3}$ as $\{\sigma_{[k]}\}_{k=1}^3$. In the **Bottom** section, we present directed interaction graphs where the outgoing edges from each node are equally weighted, and their weights collectively sum to one. The relationship between the **Top** and **Bottom** sections is established through directed edges: an edge directed from node $m$ to node $n$ with a weight of $\Sigma_{mn}$ signifies that policy $m$ is optimized against policy $n$ with a probability of $\Sigma_{mn}$. Note that this figure is reproduced from Liu et al. (2022c) and this concept is initially proposed by Garnelo et al. (2021).

### 3.2.1 Integration into Our Framework

Traditional self-play algorithms can be incorporated into our proposed framework (Algo. 1) with the following settings. **First**, the policy population $\Pi$ utilizes lazy initialization because the policy population in traditional self-play algorithms is intended to grow with each iteration. **Second**, we set $E = 1$ because only the *inactive* policy can be trained in each iteration, then turning into an *active* policy. Here, the policy population size $K$ serves as the upper limit for the number of *active* policies in the population. In other words, we use $K$ iterations to optimize the policy. **Third**, the strategy being trained $\pi_{[k]}^h$ can be initialized in a general manner. For instance, the strategy can be initialized randomly, learning from scratch. More often, $\pi_{[k]}^h$ is initialized by $\pi_{[k-1]}(\cdot | h(k-1))$, allowing incremental learning and adaptation based on the most current trained policy to accelerate convergence. **Fourth**, the policies in traditional self-play algorithms don't need supplementary information, we set the policy condition function $h(k) = \emptyset$. **Fifth**, the MSSs of traditional self-play algorithms are straightforward, often yielding a constant interaction matrix $\Sigma$ that eliminates performance evaluation. Only the MSS for prioritized fictitious self-play (described in Sec. 3.2.5) requires the performance matrix; nonetheless, it does not need to solve for complex game outcomes like NE.

Next, we outline the traditional self-play schemes. For simplicity, we operate under the following assumption:

**Assumption 1.** *In a two-player symmetric game, $C_1 = K$, with $\Sigma_{mn}$ denoting the probability that policy $m$ is optimized in response to policy $n$ which leads to $\sum_{n=1}^K \Sigma_{mn} = 1$, $\forall m$.*

Based on Assumption 1, we can further deduce the following important corollary:

**Corollary 1.** *In traditional self-play algorithms, the interaction matrix $\Sigma$ is a lower triangular matrix.*

*Proof.* The policy population gradually increases over time. When policy $m$ is selected for training, only policy $n$ ($n \leq m$) has already been trained and holds meaningful outcomes. Other policies are *inactive* policies. As a result, we exclusively select policy $n$, where $n \leq m$, to serve as the opponent for policy $m$. ☐

### 3.2.2 Vanilla Self-Play

In vanilla self-play (Samuel, 1959), agents are trained by competing against their latest versions. Thus, the MSS of vanilla self-play:

$$MSS(\mathcal{P})_{mn} = \left\{ \begin{array}{l} 1, \text{ if } n = m - 1 \\ 0, \text{ otherwise} \end{array} \right. . \tag{5}$$

Regardless of $\mathcal{P}$, the MSS produces the same interaction matrix. Although this MSS is straightforward, it is utilized in many other algorithms, so we refer to it as **vanilla MSS**. Vanilla self-play is effective in transitive games, but it can lead the agent to cyclic learning patterns in non-transitive games. It may also lead to a nonstationary problem, causing the system to get stuck in a local optimum.

A more modern version of vanilla self-play is to combine it with off-policy RL. In each iteration, the policy to be trained, $\pi_{[k]}^h$, is initialized using the previous policy $\pi_{[k-1]}(\cdot|h(k-1))$. Typically, each player's policy is fixed, and samples are collected into a buffer by vanilla self-play. These samples remain useful for training, even if they were generated by an earlier policy. This allows the agent to leverage past experiences and maintain more stationary training. Once the policy is updated, it will be added to the policy population. Moreover, vanilla self-play is frequently used in conjunction with Monte Carlo Tree Search (MCTS), and we provide a detailed discussion in Sec. 4.1.1, using the game of Go as a representative application.

### 3.2.3 Fictitious Play

Fictitious Play (FP) (Brown, 1951) is a learning algorithm in game theory where each player best responds to the empirical frequency of the strategies used by the opponent. If the opponent's strategy is static, FP can find the NE. Based on FP intuition, Fictitious Self-Play (FSP) (Heinrich et al., 2015) is introduced to make agents play against past versions of themselves to learn optimal strategies to improve the robustness of vanilla self-play. Neural Fictitious Self-Play (NFSP) (Heinrich & Silver, 2016) is a modern variant that combines FSP with deep learning techniques. It uses neural networks to approximate the BRs. Both NFSP and FSP utilize two distinct types of memory: one to store the agent's own behavioral data, and the other to record observed opponent behavior.

More recent approaches (Lanctot et al., 2017) adopt FP in meta-games, where the opponent's average strategy is approximated directly, eliminating the need to maintain two separate types of memory. In this setting, the MSS of FP is defined as:

$$MSS(\mathcal{P})_{mn} = \left\{ \begin{array}{l} \frac{1}{m-1}, \text{ if } n \leq m - 1 \\ 0, \text{ otherwise} \end{array} \right. . \tag{6}$$

In FP, the MSS continues to generate a constant interaction matrix. Compared to vanilla self-play, this approach enhances the robustness of the policy by sampling older versions of its own policies to be used as opponent strategies.

### 3.2.4 $\delta$-Uniform Self-Play

$\delta$-uniform self-play, introduced by Bansal et al. (2018), uses the hyper-parameter $\delta$, ranging from 0 to 1, to select the most recent $1 - \delta$ percentage of policies for uniform sampling to generate opponent policies. When policy $m$ is in the training phase, according to Corollary 1, only the previous $m - 1$ policies are relevant. To select opponents for policy $m$, we sample from the range of policies between $[\lceil \delta(m-1) \rceil, m-1]$, where $\lceil \cdot \rceil$

denotes the ceiling function, which rounds up the input to the nearest integer greater than or equal to it. When $\delta = 0$, the system retains the complete historical memory, whereas $\delta = 1$ implies that only the most recent policy is utilized. Thus, the MSS of $\delta$-uniform self-play:

$$MSS(\mathcal{P})_{mn} = \begin{cases} \frac{1}{f(m)}, & \text{if } \lceil \delta(m-1) \rceil \leq n \leq m-1 \\ 0, & \text{otherwise} \end{cases}, \tag{7}$$

$$f(m) = m - \lceil \delta(m-1) \rceil. \tag{8}$$

In $\delta$-uniform self-play, The MSS generates a constant interaction matrix. *Specially*, if $\delta = 0$, it corresponds to FP, and if $\delta = 1$, it corresponds to vanilla self-play.

### 3.2.5  Prioritized Fictitious Self-Play

Prioritized Fictitious Self-Play (PFSP)  (Vinyals et al., 2019) leverages a priority function to allocate a higher probability of selection to agents with higher priorities. Here, $\mathcal{P}$ represents the winning rates, specifically defined as $\mathcal{P}_{mn} = \text{Prob}(\pi_{[m]} \text{ beats } \pi_{[n]})$. The MSS of PFSP is given by Algo. 5. The function $f : [0, 1] \rightarrow [0, \infty)$ is a priority function. For example, $f(x) = (1 - x)^p$ with $p > 0$ indicates that stronger policies against the currently being trained policy have a higher chance of being chosen as opponents. Alternatively, $f = x(1 - x)$ implies that players of similar levels are more likely to be chosen as opponents. Additionally, in broader terms, $\mathcal{P}$ can also be assessed by other metrics. A similar MSS in  Berner et al. (2019) can be utilized to allocate higher probabilities to strategies that perform better or are more challenging to defeat.

---

**Algorithm 5** PFSP meta-strategy solver.

---

1: **function** $\mathcal{F}(\mathcal{P})$
2:    $\sigma_{m+1,n} \leftarrow \frac{f(\mathcal{P}_{m,n})}{\sum_{n \leq m} f(\mathcal{P}_{m,n})}$, if $n \leq m$
3:    Append zeros to $\sigma_{m+1}$ until its length is K.
4: **return** $\Sigma$

---

### 3.3  PSRO Series

Similar to traditional self-play algorithms, the PSRO series of algorithms starts with a single policy and gradually expands the policy space by incorporating new oracles. These oracles are policies that approximate optimal responses to the current meta-strategies of other agents.

### 3.3.1  Integration into Our Framework

The PSRO series of algorithms can also be integrated into our proposed framework (Algo. 1). **First**, similar to traditional self-play algorithms, we also utilize lazy initialization to initialize $\Pi$. **Second**, we also set $E = 1$ and $K$ can be considered as the upper limit for the policy population size in the original PSRO algorithms. **Third**, in the context of the PSRO series of algorithms, the strategy of our player $\pi_{[k]}^h$ can also be initialized in a general manner. **Fourth**, we set $h(k) = \emptyset$ since the PSRO series of algorithms do not use any policy condition function for their policies. **Fifth**, it's crucial to highlight that our framework diverges from the traditional PSRO model (Lanctot et al., 2017) in how $\sigma_{[k]}$ is defined. In contrast to being the meta-strategy for policy, in our framework, $\sigma_{[k]}$ is the opponent sampling strategy. It means that $\sigma_{[k]}$ here represents the opponent's meta-strategy against policy $k$ for the PSRO series of algorithms. **Sixth**, compared with traditional self-play methods, the MSSs of the PSRO series are often more complex. For example, some MSSs incorporate concepts from different types of game equilibria (McMahan et al., 2003; Muller et al., 2020; Marris et al., 2021). **Lastly**, we also simply follow Assumption 1. We can derive the Corollary 2 using a similar proof as Corollary 1.

**Corollary 2.** *In the PSRO series of algorithms, the interaction matrix $\Sigma$ is a lower triangular matrix.*

### 3.3.2 Double Oracle

Double Oracle (DO) (McMahan et al., 2003) can be only applied to two-player normal-form games. In this context, we can utilize the payoff matrix as the evaluation matrix. The interaction matrix can be initialized with all zeros, reflecting the initial absence of interactions between strategies. The MSS of DO can then be outlined as described in Algo. 6. The opponent sampling strategy $\sigma_{[k]}$ corresponds to the opponent's NE strategy of the restricted game. Therefore, the oracle in DO is a BR rather than an ABR, computing the BR against the current NE opponent strategy of the restricted game. In the context of two-player normal-form games, DO theoretically can achieve the NE of the full game.

---

**Algorithm 6** NE-based meta-strategy solver.

---

1: **function** $\mathcal{F}(\mathcal{P})$
2:    $\sigma_{[k+1]} \leftarrow$ SOLVE-NASH$(\mathcal{P}_{1:k,1:k})$ ▷ Opponent's NE meta-strategy.
3:    Append zeros to $\sigma_{[k+1]}$ until its length is K.
4: **return** $\Sigma$

---

We take *rock-paper-scissors* as an example, a symmetric two-player game. The policy condition function $h(k) = \emptyset$. The policy population size $K = 3$. The policy population $\Pi$ uses lazy initialization, and the interaction matrix $\Sigma$ is initialized with all zeros. Assume that in the **first** iteration, the policy $\pi_1$ is set to *rock*. Obviously, the NE-based MSS returns $\sigma_2 = (1, 0, 0)$. In the **second** iteration, the BR oracle adds *paper* to $\Pi$ as $\pi_2$. Consequently, the NE-based MSS yields $\sigma_3 = (0, 1, 0)$, since *paper* always beats *rock*. In the **third** iteration, noting that $\sigma_3$ selects only *paper* and never *rock*, the BR oracle adds *scissors* as $\pi_3$. The final interaction matrix is:

$$\Sigma = \begin{bmatrix} 0 & 0 & 0 \\ 1 & 0 & 0 \\ 0 & 1 & 0 \end{bmatrix} \tag{9}$$

Finally, the NE over this policy population becomes the uniform distribution over *rock*, *paper*, and *scissors*, indicating convergence. This process demonstrates how the DO algorithm iteratively expands the strategy set within our framework. Also, using DO as an example, we show that, compared with traditional self-play algorithms, the PSRO series employs more sophisticated MSSs to better capture the essence of the game, albeit at a higher computational cost.

### 3.3.3 PSRO

PSRO (Lanctot et al., 2017) extends DO to more complex games beyond just two-player normal-form games. This approach first introduces the concept of the MSS, which is a broader concept than simply computing NE. The MSS framework allows for a more flexible representation of strategic solutions in various game settings. Many variants of PSRO focus on designing new MSSs to better capture different aspects of strategic play in these more complex games. In the original PSRO framework, the oracle is computed using RL techniques, similar to those described in Algorithm 2. This allows the algorithm to effectively handle large and intricate strategy spaces, making it applicable to various game scenarios.

### 3.3.4 PSRO Variants

The traditional PSRO algorithm has inspired numerous extensions in recent research. Although the original PSRO aligns well with our proposed framework, its variants may fall outside its scope; however, they remain noteworthy and merit discussion. The first line of PSRO variants focuses on accelerating convergence. By establishing a hierarchical structure of RL workers, Pipeline PSRO (McAleer et al., 2020) achieves the parallelization of PSRO and simultaneously provides guarantees of convergence. Efficient PSRO (Zhou et al., 2022) introduces the unrestricted-restricted (URR) game to narrow the selection of opponent policies, thereby preventing the need for meta-game simulations in traditional PSRO. Similar to Pipeline PSRO, Efficient PSRO is equipped to solve URR in parallel. In addition, unlike traditional PSRO, which confines the integration of population strategies to the game's initial state, Extensive-Form Double Oracle (XDO) (McAleer et al., 2021) allows this integration across all information states. It ensures a linear convergence to an approximate

NE based on the number of information states, enhancing its tractability in extensive-form games. Online Double Oracle (ODO) (Dinh et al., 2022) integrates the no-regret analysis of online learning with DO to improve both the convergence rate to an NE and the average payoff. Anytime PSRO (McAleer et al., 2022b) and Self-play PSRO (McAleer et al., 2022a) are designed to incorporate less exploitable policies into the policy population, facilitating faster convergence. Smith et al. (2021) proposes Mixed-Oracles and Mixed-Opponents, which modify how new policies are added to the empirical game by training responses to a single opponent policy, thereby reducing the cost of simulation.

In parallel, other works aim to make PSRO more computationally tractable in different game settings. As the number of agents grows, determining BRs becomes exponentially challenging. Mean-field PSRO (Muller et al., 2022) has been introduced to address this complexity in mean-field games. In addition, due to the computational intractability of solving NE in multi-player general-sum games (Daskalakis et al., 2009; Daskalakis, 2013) and the selection problem in NE (Goldberg et al., 2013), Muller et al. (2020) put forward $\alpha$-PSRO. Instead of NE, $\alpha$-rank (Omidshafiei et al., 2019), which is unique and polynomial-time solvable in multi-player general-sum games, is introduced in the MSS, and a preference-based best response (PBR) oracle is incorporated in the approach. Similar to $\alpha$-PSRO, Marris et al. (2021) proposes the Joint Policy-Space Response Oracle (JPSRO) to tackle multi-player general-sum games. It utilizes CE and coarse correlated equilibrium (CCE) as alternatives to NE to make it computationally tractable in multi-player general-sum games and designs the CE and CCE-based MSS. Li et al. (2023) and Li et al. (2024b) enhance the scalability of PSRO for large-scale pursuit-evasion games by incorporating pre-training and fine-tuning paradigms into the policy learning process.

Other lines of research focus on promoting policy diversity, especially in non-transitive games where a single policy is often insufficient. Balduzzi et al. (2019) introduces an open-ended framework in two-player zero-sum games. This framework enhances the diversity of the strategy population and introduces a gamescape, which geometrically represents the latent objectives in games for open-ended learning. The study proposed two algorithms: Nash response $PSRO_N$ and rectified Nash response $PSRO_{rN}$. Both algorithms utilize an asymmetric payoff matrix as their performance evaluation metric. Similarly to DO, they employ the Nash-based MSS (Algo. 6). Compared to $PSRO_N$, $PSRO_{rN}$ incorporates an additional step within the ABR oracle to focus on the opponents they beat or tie and ignore the opponents they lose to. Perez-Nieves et al. (2021) uses determinantal point process (Kulesza et al., 2012) to assess diversity and introduces diverse PSRO by incorporating a diversity term into the PSRO oracle. This modification can also be easily implemented in FP and $\alpha$-PSRO. Similarly, Liu et al. (2021) introduces the concepts of behavioral diversity and response diversity and incorporates them into the PSRO oracle. Policy Space Diversity PSRO (Yao et al., 2023) defines a diversity metric named population exploitability that helps to achieve a full-game NE.

### 3.3.5 R-NaD

Although R-NaD (Perolat et al., 2021) is initially described as leveraging evolutionary game theory with a regularization component. Here, we categorized it into the PSRO series with a unique oracle computation technique (Algo. 3), executed in two stages: the first stage involves transforming the reward based on the regularization policy to make it policy-dependent, and the second stage applies replicator dynamics (Taylor & Jonker, 1978) until convergence to a fixed point. The oracle added to the policy population $\Pi$ in each iteration is derived from the reward-transformed game, not the original problem's oracle. Nonetheless, this approach ensures that the policy converges to the NE of the original game when the game is monotone. The MSS of R-NaD is the vanilla MSS, as described in Equ. (5). This equation illustrates that the fixed point reached in each iteration, the oracle, is utilized as the regularization policy for the next iteration.

### 3.4 Ongoing-Training-Based Series

In the PSRO series of algorithms, two key challenges arise. First, truncating the ABR operators during each iteration is often necessary when operating with a limited budget, introducing sub-optimally trained responses into the population. Second, relearning basic skills in every iteration is redundant and becomes untenable when confronted with increasingly formidable opponents (Liu et al., 2022c). To address these challenges, the ongoing-training-based series of algorithms emphasizes the repeated ongoing training of all

policies. In other words, instead of only *inactive* policies being selected for training, all *active* policies are also likely to be chosen for training.

### 3.4.1 Integration into Our Framework

We can incorporate these ongoing-training-based series of algorithms into our proposed framework (Algo. 1) using the following settings: **First**, we use immediate initialization to initialize $\Pi$ because, in the ongoing-training-based series, all policies in the policy population are trained together, rather than the policy population growing with each iteration. **Second**, we set $E = E_{max} > 1$, which represents the maximum number of epochs to optimize each policy within the policy population. In other words, each unique policy undergoes iterative training for $E_{max}$ times. **Third**, since each policy undergoes training for $E_{max}$ times, we utilize $\pi_{[k]}(\cdot|h(k))$ to initialize $\pi_{[k]}^h$. This means that policy updates are self-referential. **Lastly**, under Assumption 1, different from Collory 1 and 2, due to the continuous training process of all policies, we derive Collory 3.

**Corollary 3.** *In the ongoing-training-based series of algorithms, the interaction matrix $\Sigma$ is generally **not** a lower triangular matrix.*

*Proof.* When policy $m$ is selected for training, policy $n$ ($n \geq m$) has already been actually initialized and is therefore considered an *active* policy. Furthermore, in epoch $e$ ($e > 1$), policy $n$ has already been updated and holds significant meaning. As a result, policy $n$, where $n \geq m$, is likely to be chosen as the opponent for policy $m$. Consequently, the interaction matrix $\Sigma$ is generally **not** a lower triangular matrix. $\qquad\square$

### 3.4.2 FTW

Quake III Arena Capture the Flag is a renowned 3D multi-player first-person video game where two teams vie to capture as many flags as possible. The For The Win (FTW) agent (Jaderberg et al., 2019) is designed to perform at human-level proficiency in this game. A pivotal aspect of FTW is its employment of the ongoing-training-based self-play in RL. Specifically, it trains a population of $K$ different policies in parallel, which compete and collaborate with each other. When policy $k$ is undergoing training, FTW samples both its teammates and adversaries from the population. *Specially*, in scenarios where each team comprises a single member, it can be seamlessly integrated into our framework using the subsequent MSS:

$$MSS(\mathcal{P})_{mn} = \frac{1}{K}. \tag{10}$$

This essentially means that the interaction graph is densely connected. Moreover, all policies draw upon a unified policy network parameterized by $\phi$. Hence, $\pi_{[k]}(\cdot|h(k))$ can be aptly depicted as $\pi_\phi(\cdot|h(k))$. Furthermore, since these policies are not conditioned on external parameters, it is straightforward to represent the conditioning function $h(k) = \emptyset$.

### 3.4.3 NeuPL

Neural Population Learning (NeuPL) (Liu et al., 2022c) introduces another critical innovation: it employs a unified conditional network, where each policy is adjusted against a specific meta-game mixture strategy. This is instrumental in enabling transfer learning across policies. Owing to NeuPL's reliance on a unified conditional network parameterized by $\theta$, $\pi_{[k]}(\cdot|h(k))$ can be succinctly represented as $\pi_\theta(\cdot|h(k))$. Since NeuPL policies depend on the opponent sampling strategy $\sigma_{[k]}$, we define $h(k) = \sigma_{[k]}$.

### 3.4.4 Simplex-NeuPL

Simplex-NeuPL (Liu et al., 2022a), which builds on NeuPL, is designed to achieve *any-mixture optimality*, which signifies that the formulated policy should exhibit flexibility across a diverse spectrum of adversaries, including those that might not present equivalent competitive prowess. To model the population learning process from a geometric perspective, Simplex-NeuPL introduces the concept of the population simplex. Analogously to its predecessor, Simplex-NeuPL integrates a conditional network to characterize the policy,

represented as $\pi_\theta(\cdot|h(k))$ conditioned on the opponent sampling strategy $h(k) = \sigma_{[k]}$. Intriguingly, there is a slight possibility that $\sigma_{[k]}$ does not originate from the MSS. Instead, it is drawn as a sample from the population simplex. This sampling mechanism results in greater robustness.

### 3.5 Regret-Minimization-Based Series

Another line of self-play algorithms is based on regret minimization (RM). Unlike other categories, this series aims to minimize cumulative regret over time, a concept rooted in online learning theory. This distinction is crucial in two-player zero-sum games, where these methods enjoy strong theoretical guarantees. Traditional regret-minimization-based self-play typically doesn't use RL, but later studies have combined them for improved performance. This section will also discuss traditional regret-minimization methods to lay the groundwork for understanding their integration with RL.

#### 3.5.1 Integration into Our Framework

We can also integrate the regret-minimization-based series of algorithms into our proposed framework (Algo. 1) with the following settings: **First**, similar to traditional self-play algorithms and the PSRO series, we use lazy initialization to initialize the policy population $\Pi$. **Second**, we set $E = 1$, and $K$ is regarded as the maximum iteration to optimize the policy. **Third**, in each iteration $k$, $h(k)$ represents the specific elements that regret-minimization-based self-play algorithms need to store. Note that this category relies heavily on the information in $h(k)$. For instance, in Counterfactual Regret Minimization (CFR) (Zinkevich et al., 2007), it is necessary to store counterfactual regrets for every action in every information set for each player within $h(k)$. Once $h(k)$ is determined, the corresponding policy is also defined. We will discuss CFR in detail in Sec. 3.5.2. **Fourth**, we initialize $\pi_{[k]}^h$ using $\pi_{[k-1]}(\cdot|h(k-1))$ to utilize the most current trained policy. More specifically, $h(k)$ is initialized by $h(k-1)$ and $\pi_{[k]}^h$ is initialized by $\pi_{[k-1]}^h$. In each new iteration, regret-minimization information is continuously added to $h(k)$ (Line 4 in Algo. 4). As a result, $h$ accumulates information across all iterations until the end of training. **Fifth**, the ABR operator (Algo. 4) incorporates regret-related information into $h(k)$. Unlike the original CFR, Algo. 4 updates the regrets of each player sequentially. This means that the regrets of player 2 are updated after considering the already-updated regrets of player 1. This adjustment has been shown to accelerate convergence empirically and possesses a theoretical error bound (Burch et al., 2019). Additionally, each $\pi_{[k]}^h$ represents the strategies for all players, and in iteration $k$, player $i$ uses the strategy $\pi_{[k]}^h(i)$. **Lastly**, the MSS of the regret-minimization-based series is the vanilla MSS, as described in Equ. (5). Also, we can derive Collory 4.

**Corollary 4.** *In the regret-minimization-based series of algorithms, the interaction matrix $\Sigma$ is a lower triangular matrix. More specifically, it is a **unit lower shift matrix**, with ones only on the subdiagonal and zeroes elsewhere.*

*Proof.* Regret minimization-based algorithms always use the latest strategy for training. In other words, at iteration $k$, $\pi_{[k-1]}^h$ is consistently chosen as the opponent policy. Consequently, the interaction matrix $\Sigma$ is a unit lower shift matrix. $\qquad\square$

#### 3.5.2 Vanilla CFR

Regret measures the difference between the best possible payoff and the actual payoff. The regret matching algorithm (Hart & Mas-Colell, 2000) optimizes decisions by selecting strategies based on accumulated positive **overall regrets**, with higher overall regret strategies being more likely chosen to correct past underperformance. After each round, players update the overall regret values for each strategy. The algorithm is guaranteed to converge to the set of NE, and any finite stopping point yields an approximate NE. However, this method primarily applies to normal-form games because computing overall regret in extensive-form games is challenging.

Zinkevich et al. (2007) proposes CFR for extensive-form games. In this survey, we refer to it as vanilla CFR to distinguish it from later advancements. Vanilla CFR maintains both the strategy and the counterfactual regret values for each information set. Theoretically, the sum of these local regrets provides an upper bound

on the overall regret. Therefore, the problem can be decomposed into many smaller regret minimization subproblems. CFR focuses on minimizing regret locally at each information set, making the approach computationally tractable. At each iteration, the strategy is updated using regret matching, and the final strategy is obtained through strategy averaging, where the strategies played across iterations are averaged over time. This averaged strategy is proven to converge to an NE in two-player zero-sum games.

Vanilla CFR has several shortcomings. Firstly, it requires traversing the entire game tree in each iteration, computationally intractable for larger game trees. Although some efforts have focused on game abstraction to reduce the size of the game tree, greater abstraction can lead to decreased performance. Second, it requires storing counterfactual regrets $R^k(I, a)$ for every action $a$ in every information $I$ at each iteration $k$. These values are stored in $h(k)$ within our proposed framework (Algo. 1), leading to significant storage challenges.

### 3.5.3 Time-Saving Variants of CFR

Many studies focus on enhancing the time efficiency of CFR. The first approach involves modifying the regret calculation to increase its speed. CFR+ (Tammelin, 2014) implements regret-matching+ by storing non-negative regret-like values $R^{+,k}(I, a)$, rather than $R^k(I, a)$ in $k^{th}$ iteration. Additionally, CFR+ updates the regrets of each player sequentially and adopts a weighted average strategy. Moreover, Brown & Sandholm (2019a) introduces the concept of weighted regrets and develops Linear CFR (LCFR) and Discounted CFR (DCFR). The second approach involves adopting sampling methods. While it requires more iterations, each iteration is shorter, reducing the overall convergence time. Monte Carlo CFR (MCCFR) (Lanctot et al., 2009) divides terminal histories into blocks, with each iteration sampling from these blocks instead of traversing the entire game tree. This allows for calculating sampled counterfactual values for each player, leading to counterfactual regrets that, in expectation, match those of the Vanilla CFR. Vanilla CFR is a specific case where all histories are divided into just one block. MCCFR typically manifests in three forms: outcome-sampling MCCFR, where each block corresponds to a single history; external-sampling MCCFR, which samples opponent and chance nodes; and chance-sampling MCCFR, only focusing on chance nodes. Moreover, Johanson et al. (2012b) expands on chance-sampling by categorizing based on the handling of public and private chance nodes. Some studies also focus on learning how to reduce the variance of MCCFR to speed up convergence (Schmid et al., 2019; Davis et al., 2020). In addition to these two primary approaches, other studies have identified that warm starting (Brown & Sandholm, 2016) and pruning (Brown & Sandholm, 2015; Brown et al., 2017; Brown & Sandholm, 2017) can also accelerate convergence.

### 3.5.4 Space-Saving Variants of CFR

In perfect information games, decomposition reduces problem-solving scale by solving subgames. However, in imperfect information games, defining subgames is challenging due to their intersection with information set boundaries. CFR-D (Burch et al., 2014) is a pioneering method for decomposing imperfect information games into the main component called the trunk and subgames defined by forests of trees that do not divide any information sets. In each iteration, CFR is applied within the trunk for both players, and a solver is used to determine the counterfactual BR in the subgame. The process includes updating the trunk with the counterfactual values from the subgame's root and updating the average counterfactual values at this root, while the solution to the subgame is discarded. CFR-D minimizes storage needs by only saving $R^k(I_*, a)$ for information sets in the trunk and at each subgame's root, which is denoted by $I_*$, trading off storage efficiency against the time required to resolve subgames. Similar thoughts are echoed in the Continue-Resolving technique used by DeepStack (Moravčík et al., 2017) and the Safe and Nested Subgame Solving technique used by Libratus (Brown & Sandholm, 2018). We will discuss these approaches in Sec. 4.2.1, exploring their application to Texas Hold'em.

### 3.5.5 Estimation Variants of CFR

Although these CFR variants advance the field, they can't directly solve large imperfect-information extensive-form games due to their reliance on tabular representations. The typical approach involves abstracting the original game, applying CFR to the abstracted game, and translating strategies back to the original. This abstraction is game-specific and relies heavily on domain knowledge. Additionally, smaller ab-

Table 1: Overview of representative self-play algorithms within our framework.

| Algorithms | MSS | h(k) | Categories | $E$ | Initialization of $\Pi$ | Initialization of $\pi^h_{[k]}$ |
|---|---|---|---|---|---|---|
| Vanilla Self-Play | Equ. (5) | | | | | |
| FP | Equ. (6) | $\emptyset$ | Traditional Self-Play | $E=1$ | Lazy | General |
| $\delta$-Uniform Self-Play | Equ. (7) | | | | | |
| PFSP | Algo. 5 | | | | | |
| DO | NE-Based (Algo. 6) | | | | | |
| PSRO | General | | | | | |
| $\alpha$-PSRO | $\alpha$-Rank-Based | $\emptyset$ | PSRO Series | $E=1$ | Lazy | General |
| JPSRO | (C)CE-Based | | | | | |
| R-NaD | Equ. (5) | | | | | |
| FTW | Equ. (10) | $\emptyset$ | | | | |
| NeuPL | General | $\sigma_{[k]}$ | Ongoing-Training-Based | $E>1$ | Immediate | $\pi^h_{[k]} \leftarrow \pi_{[k]}(\cdot\|h(k))$ |
| Simplex-NeuPL | General | $\sigma_{[k]}$ | | | | |
| Vanilla CFR | | $R^k(I,a)$ | | | | |
| CFR+ | | $R^{+,k}(I,a)$ | | | | |
| CFR-D | Equ. (5) | $R^k(I_*,a)$ | Regret-Minimization-Based | $E=1$ | Lazy | $\pi^h_{[k]} \leftarrow \pi_{[k-1]}(\cdot\|h(k-1))$ |
| RCFR | | $\varphi(I,a)$ | | | | |
| Deep CFR | | $V(I,a\|\theta_p)$ | | | | |

stractions often yield suboptimal results. Given these challenges, Waugh et al. (2015) introduces Regression CFR (RCFR), which employs a shared regressor $\varphi(I,a)$ to estimate counterfactual regrets. Nevertheless, using regression trees as the regressor limits RCFR's applicability to miniature games, and the necessity for manually crafted features remains a drawback. After advantage-based regret minimization (ARM) (Jin et al., 2018) merges CFR with deep RL in single-agent scenarios, a growing body of research has focused on applying CFR in conjunction with neural networks to multi-agent scenarios. Double Neural CFR (Li et al., 2020a) utilizes two neural networks: one for estimating counterfactual regrets and another for approximating the average strategy. In a similar vein, Deep CFR (Brown et al., 2019) leverages an advantage network $V(I,a\|\theta_p)$ to estimate counterfactual regrets with each player having a distinct hyperparameter $\theta_p$ and employs $\pi(I,a\|\theta_\pi)$ for strategy estimation after the training process of the advantage network. Since these two networks are trained in sequence rather than concurrently, the strategy for each intermediate iteration remains conditioned on the output of the advantage network: $h(k) = V(I,a\|\theta_p)$. Despite similarities, Deep CFR distinguishes itself from Double Neural CFR through its data collection and proven effectiveness in larger-scale poker games. Single Deep CFR (SD-CFR) (Steinberger, 2019) demonstrates training an average strategy network is unnecessary, with only an advantage network required. Building on the foundation of SD-CFR, DREAM (Steinberger et al., 2020) utilizes a learned baseline to maintain low variance in a model-free setting when only one action is sampled at each decision point. Moreover, advantage regret-matching actor-critic (ARMAC) (Gruslys et al., 2020) incorporates the thought of retrospective policy improvement.

### 3.6 Reassessment of the Framework

After introducing these four categories of self-play algorithms, we will further compare them in this section, explain the rationale behind categorizing the algorithms, and summarize the representative algorithms in Table 1. Moreover, we will illustrate the differences between our framework and other frameworks to demonstrate why our proposed framework is more general. Finally, we discuss how our framework can guide the development of future self-play algorithms by offering a unified perspective on their design and evaluation.

### 3.6.1 Categorization Rationale

Traditional self-play algorithms and the PSRO series share many similarities. Initially, they require only one randomly initiated policy, and the policy population expands as training progresses. Therefore, in our

framework, we use lazy initialization to initialize the policy population and set $E = 1$ for these two categories. The interaction matrix is typically a lower triangular matrix (Corollary 1 and Corollary 2). The primary difference between the PSRO series and traditional self-play algorithms is that the PSRO series employs more complex MSSs to handle tasks with intricate game-theoretical requirements. For example, $\alpha$-PSRO (Muller et al., 2020) specifically utilizes an $\alpha$-rank-based MSS to tackle multi-player general-sum games. In other tasks, traditional self-play algorithms are more commonly used to reduce the computational cost associated with complex MSSs in the PSRO series.

Unlike the two previously mentioned categories, the ongoing-training-based series adopts a different paradigm. Instead of gradually expanding the policy population and relying on newer policies to be stronger, this approach strengthens all policies simultaneously at each epoch. This method alleviates issues such as early truncation and repeated skill relearning that occur in the above two categories. To integrate this category into our framework, immediate initialization is used for the policy population, and $\pi_{[k]}(\cdot|h(k))$ is utilized to initialize $\pi_{[k]}^h$ to ensure that policy updates are self-referential. Also, the interaction matrix is generally not a lower triangular matrix (Corollary 3).

Lastly, the regret-minimization-based series follows an online learning paradigm that focuses on optimizing overall performance across time rather than on individual episodes. Representative methods such as CFR and its variants are specifically designed for extensive-form imperfect information games, making them particularly effective in such settings. The core mechanism of regret-minimization-based training is to iteratively update the regrets associated with different strategies. However, tracking and accumulating regret for every information set incurs substantial memory overhead, especially in large game trees. Furthermore, available theoretical guarantees of convergence to a Nash equilibrium have so far been established only for two-player zero-sum games; achieving comparable assurances in multiplayer or general-sum settings remains an open research problem. Our framework uses $h(k)$ to store this information. Since the policies are determined by $h(k)$, only the most recent policy is relevant. Therefore, the interaction matrix is a unit lower shift matrix (Corollary 4). We also do not need to actually initialize the whole policy population and only need to use $\pi_{[k-1]}(\cdot|h(k-1))$ to initialize $\pi_{[k]}^h$ in the training process.

### 3.6.2 Comparing with Existing Frameworks

Our framework is built upon PSRO (Lanctot et al., 2017) and NeuPL (Liu et al., 2022c). Here, we outline the differences between our framework and these existing frameworks. The primary distinction between our framework and PSRO is the use of an interaction matrix $\Sigma := \{\sigma_{[k]}\}_{k=1}^K \in \mathbb{R}^{K \times C_1}$ to represent the opponent sampling strategy, allowing for the integration of more complex competitive dynamics. Moreover, in our framework, $\sigma_{[k]}$ denotes the opponent sampling strategy, which specifies how to sample opponents' policies against policy $k$ rather than being the meta-strategy of policy $k$. Additionally, our framework incorporates a policy condition function $h(k)$, making it more general than NeuPL, where policies are conditioned on $\sigma_{[k]}$. This enhancement gives our framework greater expressiveness. Furthermore, we describe how to compute the oracle (Line 5 in Alg. 1) in three different ways (Alg. 2, Alg. 3 and Alg. 4) to provide a clearer understanding. Also, to the best of our knowledge, our framework is the first self-play framework to integrate the regret-minimization-based series, which is a significant self-play paradigm.

### 3.6.3 Implications for Future Self-Play Algorithm Design

Beyond merely organizing existing algorithms, our framework provides insights into the design of future self-play methods. First, the four algorithmic categories we identify are not isolated; mechanisms developed in one family can be transplanted into another. Second, although most previous work has concentrated on the *ORACLE* and *MSS*, our unified framework uncovers additional, under-explored dimensions that warrant further investigation. Accordingly, the remainder of this section first illustrates how techniques can migrate across categories and then presents two specific research directions that arise when the full framework design space is considered.

**Cross-category inspiration.** Recent advances in self-play have explicitly blended techniques across different algorithmic categories, yielding hybrid methods that transcend traditional boundaries. RM-BR (Johanson et al., 2012a) pairs RM for one player with a BR oracle for the other, leveraging RM's equilibrium

guarantees and exploitative strength. Anytime PSRO (McAleer et al., 2022b) proposes RM-BR DO by incorporating a regret-minimization component into DO to ensure that exploitability monotonically decreases as new strategies are added. ODO (Dinh et al., 2022) fuses another online no-regret learning method multiplicative weights update (MWU) (Freund & Schapire, 1999) with DO. Our framework naturally subsumes ODO. Policies are still added iteratively, and the auxiliary information $h(k)$ records the average regret-based losses. MWU acts as the *ORACLE*. Crucially, unlike the original DO, this MWU oracle need not compute a full Nash equilibrium, reducing computational cost; yet ODO still provably converges to a Nash equilibrium in two-player zero-sum normal-form games while preserving rationality, allowing agents to exploit sub-optimal opponents. Taken together, these studies show that innovation thrives when regret-minimization-based series intersect with PSRO series, and they suggest that integrating other categories could be just as fruitful.

**Prioritised re-training when $E \gg 1$.** When each policy is reused across many encounters ($E \gg 1$), repeatedly updating all members becomes computationally expensive. A natural open question is: *which policies are worth further improvement*? For instance, we can formalize this as a multi-armed bandit: give every policy a score that estimates how much new information an additional update would add (e.g., its expected information gain), and let that score decide where to spend oracle calls. This bandit scheduling automatically balances exploitation (further refining the strongest policies) with exploration (re-examining under-trained or atypical policies that might expose blind spots).

**Auxiliary information $h(k)$ and knowledge transfer.** The history kernel $h(k)$ need not serve merely as a passive log of past interactions; instead, it can be designed to encode rich auxiliary domain knowledge, such as exploitable behavioral patterns, partial payoff structures, or latent embeddings of opponents. By treating $h(k)$ as a learnable memory object that can be *inherited* or *shared* across subsets of agents, we enable a form of continual-learning self-play. In this setting, knowledge acquired by one agent can propagate throughout the population organically, without requiring intervention from an external oracle. This facilitates more efficient collective learning and strategic evolution within multi-agent systems.

## 4 Empirical Analysis

In this section, we introduce iconic applications of self-play by categorizing the scenarios into three distinct groups: board games, which typically involve perfect information; card games, which usually involve imperfect information; and video games, which feature real-time actions rather than turn-based play. We then illustrate how self-play is applied in these complex scenarios and provide a comparative analysis of these applications in Table 2.

### 4.1 Board Games

Board games, the majority of which are perfect information games, were previously revolutionized by the introduction of two essential techniques: position evaluation and MCTS (Coulom, 2006; Kocsis & Szepesvári, 2006). These methodologies, with minor modifications, demonstrated superhuman effectiveness in solving board games such as chess (Campbell et al., 2002), checkers (Schaeffer et al., 1992), othello (Buro, 1999), backgammon (Tesauro & Galperin, 1996), and Scrabble (Sheppard, 2002). In contrast, the application of these techniques to the game of Go with an estimated $2.1 \times 10^{170}$ legal board configurations, only enabled performance at the amateur level (Bouzy & Helmstetter, 2004; Coulom, 2007; Baudiš & Gailly, 2011; Enzenberger et al., 2010; Gelly & Silver, 2007). In light of this, our discussion will specifically focus on the game of Go to illustrate the application of vanilla self-play. In addition to Go, we will explore Stratego, a board game with imperfect information, in contrast to most board games that only involve perfect information.

#### 4.1.1 Go

Go is an ancient board game, which is played on a grid of 19x19 lines, where two players alternately place black and white stones aiming to control the largest territory. The paradigm of Go artificial intelligence (AI) is revolutionized with the launch of DeepMind's AlphaGo series (Silver et al., 2016; 2017; 2018; Schrittwieser et al., 2020), which leveraged the power of vanilla self-play to significantly elevate performance. In AlphaGo (Silver et al., 2016), the training regime can be split into three stages. In the first stage, supervised

learning with expert data trains a fast policy network $p_\pi(a|s)$ for rollouts in MCTS and a precise policy network $p_\sigma(a|s)$. The second stage employs vanilla self-play combined with RL to get a refined policy network $p_\rho(a|s)$ based on $p_\sigma(a|s)$ and subsequently trains a value network $v_\theta(s)$. Specifically, $p_\rho(a|s)$ is refined by competing against a randomly selected historical version $p_{\rho^-}(a|s)$, similar to the MSS shown in Equ. (6), while $v_\theta(s)$ is trained using the game samples collected by vanilla self-play of $p_\rho(a|s)$. In the third stage, MCTS integrates the policy and value networks to select actions.

Based on AlphaGo, AlphaGo Zero (Silver et al., 2017) does not require any expert data except game rules. It utilizes only one network $f_\theta(s)$ to concurrently predict the action probabilities $\Pr(a|s)$ and the state value $v$. Each move is generated by MCTS with guidance from $f_\theta(s)$ to aid MCTS expansion, unlike the rollouts used in AlphaGo. Vanilla self-play is employed to generate data and refine $f_\theta(s)$ with the current best policy competing against itself, a process analogous to the MSS referenced in Equ. (5). A new policy to be incorporated into the policy pool must surpass a 55 percent win rate against its predecessor, aligning with the stipulation set in Algo. 2 at Line 1. AlphaZero (Silver et al., 2018) extends AlphaGo Zero to include games beyond Go, such as Chess and Shogi. Notably, a draw is introduced as an additional expected outcome, and data augmentation is omitted due to the asymmetry of Chess and Shogi. Concerning the vanilla self-play procedure, the only difference between AlphaZero and AlphaGo Zero is that AlphaZero utilizes the newly updated network without the validation process present in AlphaGo Zero. Building upon AlphaZero, MuZero (Schrittwieser et al., 2020) takes learning from scratch to the next level, even operating without predefined game rules. MuZero incorporates ideas from model-based RL to model the dynamics of games. In addition to the prediction network $f$ (similar to the networks in AlphaGo Zero and AlphaZero), MuZero introduces a dynamics network $g$ to model the MDP and a representation network $h$ to map observations to hidden states. These three networks are trained jointly. MuZero employs MCTS guided by the three networks above to make decisions. The vanilla self-play process in MuZero operates similarly to that in AlphaZero. In practice, in addition to excelling in board games like Go, MuZero also achieves state-of-the-art performance in Atari games.

### 4.1.2 Stratego

Unlike most board games, which are perfect information games, Stratego, a two-player imperfect information board game, distinguishes itself by incorporating elements of memory, deduction, and bluffing. This complexity is further amplified by the game's long episode length and many potential game states, estimated $10^{535}$ (Perolat et al., 2022). The game is divided into two phases: the deployment phase, where players secretly arrange their units, and the game-play phase, where the objective is to deduce the opponent's setup and capture their flag. The depth and computational complexity of the game remained a challenge until breakthroughs such as DeepNash (Perolat et al., 2022) showed promising advances in AI's ability to tackle it. DeepNash scales up evolution-theory-based self-play method R-NaD (Perolat et al., 2021) (discussed in Sec. 3.3.5) to neural R-NaD. It employs a neural network with four heads: one for value prediction, one for the deployment phase, one for piece selection, and one for piece displacement. Neural Replicator Dynamics (Hennes et al., 2020) is utilized to obtain the approximate fixed-point policy. DeepNash holds the third-place ranking among all professional Gravon Stratego players and wins nearly every match against existing Stratego bots.

### 4.2 Card Games

Unlike board games, card games are typically imperfect information games with a larger state space and greater complexity. Some regret-minimization-based self-play methods, particularly CFR and its variants, are well-suited for handling imperfect information and have achieved strong results in card games. However, recent studies have shown that other self-play methods without regret minimization can also perform well in such settings. Here, we introduce three representative card games: Texas Hold'em, DouDiZhu, and Mahjong, to illustrate how self-play is utilized in card games.

### 4.2.1 Texas Hold'em

**Texas Hold'em**, a popular poker game with 2-10 players, is known for its strategic depth and bluffing elements. The two-player variant is called **heads-up Texas Hold'em**. The gameplay begins with each player receiving two private cards (hole cards), followed by a round of betting. Subsequently, three community cards (the flop) are revealed, leading to another betting round. This is followed by dealing a fourth (the turn) and a fifth community card (the river), each accompanied by further betting rounds. The objective is to construct the best five-card poker hand from any combination of hole cards and community cards. Texas Hold'em offers two betting formats: limited betting and no-limit betting. The latter one is noted for its complexity, allowing players to bet any amount up to their entire stack of chips. While simplified versions such as **Kuhn Poker** and **Leduc Poker** serve valuable roles in theoretical analysis, we will focus on the algorithms designed to compete in full Texas Hold'em.

**Heads-up limit Texas Hold'em (HULHE)**, the simplest form with approximately $3.16 \times 10^{17}$ game states, was not solved until the introduction of Cepheus (Bowling et al., 2015). It utilizes fixed-point arithmetic with compression and regret-minimization-based self-play method CFR+ (Tammelin, 2014) to address the issues of storage and computation, respectively, resulting in superhuman performance. Deep CFR (Brown et al., 2019) combines regret-minimization-based self-play method CFR with deep neural networks. Furthermore, some studies do not adopt the regret-minimization-based series of self-play; instead, they use the traditional self-play method. NSFP (Heinrich & Silver, 2016) introduces a self-play method combined with end-to-end RL training, also achieving competitive performance in HULHE. Poker-CNN (Yakovenko et al., 2016) utilizes self-play to learn convolutional networks to solve a video version of HULHE.

After solving HULHE, research focus shifts to **heads-up no-limit Texas Hold'em (HUNL)** with significantly larger game states, approximately $10^{164}$ (Johanson, 2013). Thus, traversing the game tree as Cepheus does in HULHE is impossible. DeepStack (Moravčík et al., 2017) is a regret-minimization-based self-play method combined with continual re-solving. This method focuses on a subtree of limited depth and breadth and estimates the outcomes of the furthest reaches of this subtree. Libratus (Brown & Sandholm, 2018) develops a blueprint strategy, leveraging an enhanced version of regret-minimization-based self-play method MCCFR (Lanctot et al., 2009). Recursive Belief-based Learning (ReBeL) (Brown et al., 2020) introduces the public belief state (PBS), transforming the imperfect-information game into a perfect-information game with continuous state space. ReBeL utilizes the regret-minimization-based self-play method CFR-D, combined with RL and search, to train both the value and policy networks. Unlike the algorithms mentioned above, AlphaHoldem (Zhao et al., 2022a) does not use regret-minimization-based self-play. Instead, it proposes a $K$-best self-play method that always selects the top $K$ agents, based on their ELO ratings (Elo & Sloan, 1978), to generate samples.

Pluribus (Brown & Sandholm, 2019b), based on Libratus, addresses **six-player no-limit Texas Hold'em**. Similar to Libratus, Pluribus utilizes a regret-minimization-based self-play method MCCFR (Lanctot et al., 2009) to develop its blueprint strategy. It conducts a depth-limited search before executing actions. Different from Libratus, Pluribus maintains a streamlined policy pool of only four strategies, assuming that its opponents might adjust their strategies during gameplay among these four strategies, which allows Pluribus to manage complexity more efficiently.

### 4.2.2 DouDiZhu

DouDizhu (a.k.a. Fight the Landlord) is a three-player strategic card game. In this game, one player takes on the role of the landlord and competes against the other two players, the peasants. The game is played in two main stages: the bidding stage and the card play stage. During the bidding stage, players vie to become the landlord. During the card play stage, players take turns playing cards in various combinations, intending to be the first to empty their hands. DouDiZhu is characterized by imperfect information, as players can only see their own cards and the cards played. The essence of the game lies in cooperation among peasants and competition between the two factions. It has an estimated $10^{76} \sim 10^{106}$ possible game states (Zha et al., 2019) and an action space comprising 27472 possible moves (Yang et al., 2022).

DeltaDou (Jiang et al., 2019) is the first algorithm to achieve expert-level performance in DouDizhu. Similar to AlphaZero (Silver et al., 2018), DeltaDou utilizes vanilla self-play to generate samples to train policy and

value networks. DouZero (Zha et al., 2021) also uses vanilla self-play to generate data. It reduces training costs by opting for a sampling method over the tree search approach. Based on DouZero, DouZero+ (Zhao et al., 2022b) incorporates opponent modeling, assisting the agent in making more informed decisions. PerfectDou (Yang et al., 2022) utilizes vanilla self-play under a Perfect-Training-Imperfect-Execution framework.

### 4.2.3 Mahjong

Mahjong has evolved into various global variants, including the famous Japanese version known as Riichi Mahjong. This game is typically played by four players who must navigate both the visible aspects of the game, such as discarded tiles, and the hidden elements, like their own hand and the unseen public tiles. The strategic depth and complexity of Mahjong pose significant challenges. Despite ongoing research, these findings have yet to reach expert human levels (Kurita & Hoki, 2020; Gao et al., 2019; Mizukami & Tsuruoka, 2015). Difficulties are navigating incomplete information, dynamically adapting strategies to multiple opponents, and contending with complex winning rules and an enormous number of possible game states, estimated at $10^{169}$ (Zha et al., 2019). Suphx (Li et al., 2020b) is recognized as one of the first algorithms to master Mahjong, achieving a performance level comparable to expert human players, precisely 10 dan on Tenhou (Tsunoda), the most popular online Mahjong platform. Initially, Suphx employs supervised learning, utilizing expert data to train its model. It then advances its capabilities through vanilla self-play combined with RL. Similarly, NAGA (Village), developed by Dwango Media Village, and LuckyJ (Tencent), designed by Tencent, have also achieved the rank of 10 dan on Tenhou. Furthermore, LuckyJ has even defeated human professional players.

## 4.3 Video Games

In contrast to traditional board games and card games, video games often feature real-time actions, long trajectories, and increased complexity stemming from a broader range of actions and observations. We illustrate representative video games showcasing self-play's impact.

### 4.3.1 StarCraft II

StarCraft is a real-time strategy (RTS) game. It has three distinct species: the Terrans, Zerg, and Protoss, each with unique units and strategic options that enhance the gameplay complexity. Renowned for its balanced gameplay, strategic depth, and intense competitiveness, the game challenges players to gather resources, construct bases, and build armies. Victory requires meticulous planning and tactical execution, with defeat occurring when a player loses all their buildings. AlphaStar (Vinyals et al., 2019) dominates the 1v1 mode competitions in StarCraft II and has defeated professional players. Its framework is similar to AlphaGo (Silver et al., 2016), initially utilizing supervised learning to train the policy with expert data. Subsequently, it uses a hierarchical self-play method combined with end-to-end RL to train the networks. More specifically, The proposed self-play method divides all the agents into three types: main agents, league exploiters, and main exploiters. It maintains a policy pool of past players that records all these types of agents. **Main agents** engage in traditional self-play algorithms like FP and PFSP, competing against main agents themselves and other agents in the policy pool. They are periodically added to the pool and never reset. **League exploiters** also use a traditional self-play method PFSP to play against all policy pool agents, added to the pool if they show a high win rate and potentially reset to expose global blind spots. **Main exploiters** only compete with main agents to improve their robustness, are added to the pool after achieving a high win rate or specific training steps, and are reset upon each addition. Among those three agent types, the main agent is the core agent and embodies the final AlphaStar strategy. However, the introduction of three agent types significantly increases the training computation. Further studies (Han et al., 2020; Wang et al., 2021; Huang et al., 2024) have enhanced the league self-play training procedure.

### 4.3.2 MOBA Games

Multiplayer Online Battle Arena (MOBA) games are a popular video game genre that blends RTS with role-playing elements. In typical MOBA games, two teams of players control their unique characters, known as heroes, and compete to destroy the opposing team's main structure, often referred to as the base. Each

hero has distinct abilities and plays a specific role within the team, such as Warrior, Tank, or Support. Managing multiple lanes and battling under the fog of war, which obscures parts of the map, are critical aspects of the gameplay.

OpenAI Five (Berner et al., 2019) defeated the world champion team in a simplified version of Dota 2, which supported only 17 heroes and excluded complex features like summons and illusions. The training process introduces a self-play method that combines two traditional self-play algorithms: with an 80% probability of engaging in vanilla self-play and a 20% probability of employing a technique similar to the PFSP used in AlphaStar (Vinyals et al., 2019). This technique selects each policy from the policy pool based on its quality score, which is continuously updated from competition results throughout training. Higher quality scores increase the likelihood of a policy being selected. OpenAI Five also requires extensive training resources.

Another notable MOBA game, especially viral in China, is Honor of Kings. The 1v1 mode is conquered by Ye et al. (2020b), which boasts a significant winning rate against top professional players. It utilizes a traditional self-play method $\delta$-uniform self-play to train the RL networks. The 5v5 mode was later mastered by Ye et al. (2020a) as well. Unlike OpenAI Five, which supported only 17 heroes, this work expands the hero pool to 40 heroes, substantially increasing the number of possible lineup combinations. It proposes a new self-play method referred to as curriculum self-play learning (CSPL). Specifically, the training process is divided into three stages. The first stage involves training fixed lineups through self-play and utilizing human data to balance the two teams to aid policy improvements. The second stage employs multi-teacher policy distillation to produce a distilled model. The final stage uses this distilled model as the initial model for another round of self-play with randomly picked lineups. This approach defeats professional player teams. The self-play generated data is also used to learn compelling lineup drafting by utilizing MCTS and neural networks (Chen et al., 2021).

### 4.3.3   Google Research Football

Google Research Football (GRF) (Kurach et al., 2020) is an open-source football simulator emphasizing high-level actions. It initially offers two scenarios: the football benchmark and the football academy with 11 specific tasks. Here, we focus exclusively on the football benchmark because it presents a more complex scenario that better demonstrates the effects of self-play. GRF is particularly challenging due to the need for cooperation among teammates and competition against opposing teams. It features long trajectories with 3000 steps per round, stochastic transitions, and sparse rewards.

WeKick (Ziyang Li, 2020) claimed victory in the GRF competition on Kaggle (Google, 2020), which simplifies the game dynamics by allowing competitors to control only one player, either the ball carrier on offense or the nearest defender on defense. It employed self-play strategies similar to those used in league training (Vinyals et al., 2019). It initializes its opponent policy pool using strategies developed through RL and Generative Adversarial Imitation Learning (GAIL) (Ho & Ermon, 2016) to facilitate the training process.

Further research delves into the full football game rather than the simplified version. Team-PSRO (McAleer et al., 2023) extends PSRO series of self-play algorithms to team games, outperforming baselines in the 4v4 version of the full GRF. In the context of the 11v11 version, where the goalkeeper is rule-based controlled, TiKick (Huang et al., 2021a) utilizes vanilla self-play to collect samples and then employs imitation learning. Fictitious Cross-Play (FXP) (Xu et al., 2023) proposes a new self-play method similar to AlphaStar (Vinyals et al., 2019). It introduces two populations: the main population and the counter population. Policies in the counter population improve solely by cross-playing with policies in the main population as opponents, while policies in the main population engage in playing with policies from both populations. FXP achieves a win rate of over 94% against TiKick. TiZero (Lin et al., 2023), a follow-up to TiKick, combines curriculum learning with FP and PFSP (Vinyals et al., 2019) to avoid expert data reliance, achieving a higher TrueSkill rating (Herbrich et al., 2006) than TiKick.

Table 2: Overview of representative studies in empirical analysis.

| Category | Game | Number of Players | Perfect Information | Number of Game States | Algorithms | Self-play Method | Search | Expert Data |
|---|---|---|---|---|---|---|---|---|
| Board Game | Chess | 2 | ✓ | $10^{45}$ | AlphaZero | Vanilla SP | ✓ | × |
| | | | | | MuZero | Vanilla SP | ✓ | × |
| | Go | 2 | ✓ | $10^{360}$ | AlphaGo | FP | ✓ | ✓ |
| | | | | | AlphaGo Zero | Vanilla SP | ✓ | × |
| | | | | | AlphaZero | Vanilla SP | ✓ | × |
| | | | | | MuZero | Vanilla SP | ✓ | × |
| | Stratego | 2 | × | $10^{535}$ | DeepNash | R-NaD | × | × |
| | HULHE | 2 | × | $10^{17}$ | Cepheus | CFR+ | ✓ | × |
| | | | | | NFSP | NFSP | × | × |
| | HUNL | 2 | × | $10^{164}$ | DeepStack | CFR+ | ✓ | × |
| | | | | | Libratus | MCCFR | ✓ | × |
| | | | | | ReBel | CFR-D / FP | ✓ | × |
| Card Game | Six-player No-limit Texas Hold'em | 6 | × | $> 10^{164}$ | Pluribus | MCCFR | ✓ | × |
| | DouDiZhu | 3 | × | $10^{76} \sim 10^{106}$ | DeltaDou | Vanilla SP | ✓ | × |
| | | | | | DouZero | Vanilla SP | × | ✓ |
| | | | | | PerfectDou | Vanilla SP | × | × |
| | Mahjong | 4 | × | $10^{169}$ | Suphx | Vanilla SP | ✓ | ✓ |
| Video Game | StarCraft II | 2 | × | | AlphaStar | FP & PFSP | × | ✓ |
| | Dota 2 | 10 | × | / | OpenAI Five | Vanilla SP & PFSP | × | × |
| | Honor of Kings | 10 | × | | MOBA AI | $\delta$-uniform SP | ✓ | ✓ |
| | Google Research Football | 22 | ✓ | | TiZero | FP & PFSP | × | × |

# 5 Open Problems and Future Work

Self-play methods have shown remarkable performance by leveraging iterative learning and adapting to complex environments. However, significant challenges and open questions remain, presenting opportunities for further research and development.

## 5.1 Theoretical Foundation

Although NE has been shown to exist in games with finite players and finite actions (Nash et al., 1950), computing NE with self-play algorithms in larger games remains challenging and consequently, many studies aim to achieve approximate NE (Li et al., 2024a). However, in some cases, even computing an approximate NE is difficult (Daskalakis, 2013). Some research has resorted to alternative solution concepts, such as CE (Marris et al., 2021) and $\alpha$-rank (Muller et al., 2020). While many successful self-play algorithms are grounded in theoretical intuition, formal game-theoretic guarantees for their effectiveness in large, imperfect-information games remain limited. For instance, approaches such as AlphaGo (Silver et al., 2016), AlphaStar (Samvelyan et al., 2019), and OpenAI Five (Berner et al., 2019) achieve empirical success. However, under realistic constraints such as finite computational resources and limited exploration, their convergence or optimality remains difficult to formally characterize within standard game-theoretic frameworks. Future work may benefit from further narrowing the gap between empirical success and theoretical understanding, either by extending existing theoretical tools to more complex settings or by designing new self-play methods with provable guarantees under realistic assumptions.

## 5.2 Non-stationarity of the Environment

In the self-play framework, the opponents are a vital component of the environment, and the strategies of the opponent players evolve as training progresses. This evolution can cause the same strategy to lead to different results over time, creating a non-stationary environment. This problem is also shared by the MARL area. Future research should aim to develop self-play algorithms that are more robust and can adapt to changing conditions. For example, incorporating opponent modeling (Zhao et al., 2022b) into self-play can help agents anticipate changes in opponent strategies and adjust their own strategies proactively, making them more robust to environmental changes.

## 5.3 Scalability and Training Efficiency

The scalability of self-play methods faces significant challenges as the number of teams and players within those teams increases. As the number of participants grows, the complexity of interactions explodes. For example, in OpenAI Five (Berner et al., 2019), the hero pool size is limited to only 17 heroes. MOBA AI (Ye et al., 2020a) extends this to a 40-hero pool with the help of curriculum learning, but it still cannot cover the entire hero pool available in the actual game. One potential solution is leveraging players' inherent connections to optimize the learning process. For instance, using graph-based models to represent and exploit the relationships between players can help manage and reduce the complexity of large-scale multi-agent environments. These scalability issues are fundamentally rooted in the limited training efficiency of self-play methods from two aspects including computation and storage. The first issue is computational efficiency, which is induced by the iterative nature of self-play where agents repeatedly play against themselves or past versions. Moreover, although forming more complex populations and competitive mechanisms (Samvelyan et al., 2019) can enhance the intensity and quality of training, it further exacerbates the demand for computational resources. Techniques such as parallel computing, distributed learning, and more efficient neural network architectures could be explored to address these challenges. The second issue is storage because self-play requires maintaining a policy pool. Even when using a shared network architecture, storing the parameters of large models can be problematic. This issue is particularly pronounced in regret-minimization-based self-play algorithms, which must store the regrets for each information set and potential action. Managing both the computational load and storage requirements is essential for improving the overall training efficiency and scalability of self-play methods.

### 5.4 With Large Language Models

With their remarkable capability and emergent generalizability, large language models (LLMs) have been regarded as a potential foundation for achieving human-level intelligence (Achiam et al., 2023), and self-play methods have been proposed to fine-tune LLMs, enhance LLMs' reasoning performance, and build LLM-based agents with strong decision-making abilities. Post-training fine-tuning (Ouyang et al., 2022; Bai et al., 2022) is a key step in aligning LLM with more desired behaviors, but it requires a huge amount of human-annotated data. To reduce reliance on human-annotated data, Self-play fIne-tuNing (SPIN) (Chen et al., 2024) introduces a self-play mechanism to generate training data using the LLM itself and fine-tuning the LLM by distinguishing self-generated responses from human-annotated data. Another line of work (Munos et al., 2024; Swamy et al., 2024; Wu et al., 2024) formulate the alignment problem as a two-player constant-sum game and use self-play methods to solve the game. Some other work (Huang et al., 2023; Yuan et al., 2024) also utilizes model-generated data to fine-tune LLMs with minimal human annotations. The idea of self-improvement has also been applied to improve the reasoning ability of LLMs. Self-Play of Adversarial Game (SPAG) (Cheng et al., 2024) observes that self-play in a two-player adversarial language game called Adversarial Taboo can boost the LLM's performance on various reasoning benchmarks. Besides improving the capability of LLMs, self-play methods have also contributed to building LLM-based agents with strong strategic abilities. A representative work is Cicero (Meta et al., 2022), which achieves human-level play in the game of Diplomacy by combining language models with an RL policy trained by self-play. Cicero uses the self-play policy to produce an intended action and prompts the language model to generate languages conditioned on the policy's intent. Another line of work (Xu et al., 2024; 2025a) also combines LLM with self-play policy but takes a different approach by first prompting the LLM to propose multiple action candidates and then using the self-play policy to produce the action distribution over these candidates. Despite recent progress, applying self-play with LLMs is still underexplored and requires further research.

### 5.5 Real-World Applications

Self-play is particularly effective in addressing some problems abstracted from real-world situations through its iterative learning approach. In the field of economics, for instance, self-play is used to enhance supervised learning models in multi-issue bargaining tasks (Lewis et al., 2017). Furthermore, self-play proves beneficial in solving combinatorial optimization problems (COPs) such as the traveling salesman problem (TSP) and the capacitated vehicle routing problem (CVRP) (Wang et al., 2024). Within the domain of traffic, self-play facilitates the development of human-like autonomous driving behaviors (Cornelisse & Vinitsky, 2024) and enables vehicles to learn negotiation strategies to merge on or off roads (Tang, 2019), although currently within 2D simulators.

Deploying self-play directly in the physical world remains challenging. Its iterative, trial-and-error nature is computationally expensive and can produce behaviors that are impractical or even unsafe outside a controlled environment. Consequently, most recent work performs self-play in simulators, and practical adoption hinges on bridging the simulation-to-reality (Sim2Real) gap. When this gap is small, self-play has already proved useful. For instance, self-play can be well employed to enhance video streaming capabilities (Huang et al., 2021b), and to address the image retargeting problem (Kajiura et al., 2020). Moreover, EvoPlay (Wang et al., 2023) leverages self-play to design protein sequences, utilizing the advanced AlphaFold2 simulator (Jumper et al., 2021) to narrow the Sim2Real gap. In heterogeneous multi-robot systems, self-play is utilized for competitive sports tasks like humanoid football (Liu et al., 2022b; Haarnoja et al., 2024), quadruped competition (Xiong et al., 2024), and multi-drone volleyball (Xu et al., 2025b; Zhang et al., 2025), with substantial efforts dedicated to Sim2Real transitions for real-world success (Gao et al., 2023). A complementary research direction is to boost the sample efficiency of self-play so that policies can be trained online, reducing or even eliminating the need for elaborate Sim2Real pipelines in real-world deployments.

## 6 Conclusion

The burgeoning field of self-play in RL has been systematically explored in this survey. The core idea of self-play, that agents iteratively refine their policies by interacting with historical or concurrent versions of

themselves or other evolving agents, has proven to be a powerful approach for developing robust strategies across various domains. Before delving into the specifics of self-play, this paper first elucidates the MARL framework and introduces the game theory background of self-play. Moreover, the paper presents a unified framework and categorizes existing self-play algorithms into four main categories: traditional self-play algorithms, the PSRO series, the ongoing-training-based series, and the regret-minimization-based series. We provide detailed explanations on how these four groups are seamlessly integrated into our proposed framework, ensuring a comprehensive understanding of their functionalities. We then compare the four categories within this common lens and draw design insights that can guide future self-play research. The transition from theory to application is underscored by a rigorous analysis of self-play's role within various scenarios, such as board games, card games, and video games. Despite the successes of self-play in many areas, challenges remain, such as convergence to suboptimal strategies and substantial computational demands. Also, future work can focus on how to integrate self-play with LLMs and achieve real-world applications. In conclusion, self-play is vital to modern RL research, offering key insights and tools for developing advanced AI systems. This survey serves as an essential guide for researchers and practitioners, paving the way for further advancements in this field.

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

# A  Abbreviations in Alphabetical Order

| | |
|---|---|
| **ABR** | Approximate Best Response |
| **AI** | Artificial Intelligence |
| **ARM** | Advantage-based Regret Minimization |
| **ARMAC** | Advantage Regret-Matching Actor-Critic |
| **BR** | Best Response |
| **CCE** | Coarse Correlated Equilibrium |
| **CE** | Correlated Equilibrium |
| **CFR** | Counterfactual Regret Minimization |
| **COP** | Combinatorial Optimization Problem |
| **CSPL** | Curriculum Self-Play Learning |
| **CVRP** | Capacitated Vehicle Routing Problem |
| **DCFR** | Discounted Counterfactual Regret Minimization |
| **DO** | Double Oracle |
| **EGTA** | Empirical Game-Theoretic Analysis |
| **FP** | Fictitious Play |
| **FSP** | Fictitious Self-Play |
| **FTW** | For The Win |

| | |
|---|---|
| **FXP** | Fictitious Cross-Play |
| **GAIL** | Generative Adversarial Imitation Learning |
| **GRF** | Google Research Football |
| **HULHE** | Heads-Up Limit Texas Hold'em |
| **HUNL** | Heads-Up No-Limit Texas Hold'em |
| **JPSRO** | Joint Policy-Space Response Oracle |
| **LCFR** | Linear Counterfactual Regret Minimization |
| **LLM** | Large Language Model |
| **MARL** | Multi-Agent Reinforcement Learning |
| **MCCFR** | Monte Carlo Counterfactual Regret Minimization |
| **MCTS** | Monte Carlo Tree Search |
| **MDP** | Markov Decision Process |
| **MG** | Markov Game |
| **ML** | Machine Learning |
| **MOBA** | Multiplayer Online Battle Arena |
| **MSS** | Meta-Strategy Solver |
| **MWU** | Multiplicative Weights Update |
| **NE** | Nash Equilibrium |
| **NeuPL** | Neural Population Learning |
| **NFSP** | Neural Fictitious Self-Play |
| **ODO** | Online Double Oracle |
| **PBR** | Preference-based Best Response |
| **PBS** | Public Belief State |
| **PFSP** | Prioritized Fictitious Self-Play |
| **POMG** | Partially Observable Markov Game |
| **PSRO** | Policy-Space Response Oracle |
| **RCFR** | Regression Counterfactual Regret Minimization |
| **ReBeL** | Recursive Belief-based Learning |
| **RL** | Reinforcement Learning |
| **RM** | Regret Minimization |
| **R-NaD** | Regularized Nash Dynamics |
| **RTS** | Real-Time Strategy |
| **SD-CFR** | Single Deep Counterfactual Regret Minimization |
| **SPAG** | Self-Play of Adversarial Game |
| **SPIN** | Self-Play fIne-tuNing |
| **TSP** | Traveling Salesman Problem |
| **URR** | UnRestricted-Restricted |
| **XDO** | Extensive-Form Double Oracle |

