# OpenReview forum: "A Survey on Self-Play Methods in Reinforcement Learning"
_TMLR — Rejected by TMLR_

### Review · Reviewer_j39o · 2025-04-03

**Summary Of Contributions:**

This paper surveys a wide variety of self-play related algorithms for analyzing multiagent systems. They introduced a general framework to describe self-play and showed how different categories of algorithms can be fitted in their framework. After surveying these algorithms, they discussed the applications of self-play and its future research directions.

**Audience:**

Yes

**Claims And Evidence:**

Yes

**Requested Changes:**

Besides the two concerns, there are some imprecise or incorrect descriptions of game theory preliminaries. For example:

1. In 2.2.1, “In a game with perfect information, only one player moves at a time. Each player comprehensively understands the current game state, the complete history of moves made, and the potential future outcomes.” First, “comprehensively understands” is vague. Second, this definition is a little bit informal. It would be better to define perfect information after defining the information sets, and then have a formal definition of perfect information in terms of information sets.

2. In 2.2.2, “The game is a zero-sum game, otherwise it is a general-sum game.” I think the key of general-sum is that the sum of the players’ payoffs is not constant. So it seems that the complementary part of general sum is constant sum, and zero sum is a special case of constant sum. It would be nice to clarify this relationship.

3. The definition of symmetric games is incomplete. The missing part is that players should have the same utility function.

4. For the last line on page 3, little pi was defined as a pure strategy at the beginning of 2.2.2, while here pi is defined as a behavior strategy, which seems inconsistent. I suggest to follow the pure-strategy and mixed-strategy definitions at the beginning of 2.2.2 and define behavior strategy separately. Then you can describe the equivalence between mixed strategy and behavior strategy under certain conditions. In addition, a typo “A” should be “a”.

5. In the next sentence “A subgame of an extensive-form game is a portion of the game that starts from a single initial node, includes all successors of any node within the subgame, and contains all nodes in the same information set as any node in the subgame.” This definition seems to be circular since a subgame is defined in terms of subgame.

6. In the Prisoner’s dilemma examples for Fig.2, it is less interesting to have or compare the simultaneous-move version with the sequential move version since Defect (C) is a dominant strategy. Maybe there are more interesting examples to illustrate the difference between static games and dynamic games.

7. For the definition of meta-games, I think meta-games are not limited to be a normal-form game. They can take any form (e.g., EFG) as long as they provide a certain abstraction of the original game.

**Minors:**
1. In Algorithm 3, line 3 is hand-waving.

2. For Eq.6, I would say it is just fictitious play applied to meta games rather than FSP.

3. Typos: activa -> active, MSSes -> MSSs

**Strengths And Weaknesses:**

**Strengths:**

1. This paper surveys a wide range of algorithms, connecting the game-theoretic community and the MARL community.

2. Examples are used to illustrate different algorithms, which help readers to understand the algorithms.

**Weaknesses:**

I have two major concerns on this paper:

First, I am deeply concerned about whether MARL and game-theoretic algorithms can be accurately categorized under the umbrella of self-play. If they are to be considered as such, a clear and very general definition of self-play is necessary—something currently missing from the paper, and for which no consensus exists. In the case of classic self-play, it seems more accurate to view it as a special case of certain algorithms (e.g., PSRO), rather than considering PSRO as an instance of self-play. Even if a general definition were provided, I remain skeptical about whether such a definition is actually useful or necessary for the purposes of classification.

Second, the self-play framework presented in the paper appears to subsume existing algorithms in a rather brute-force fashion. However, the paper does not clearly articulate the significance of this abstraction—specifically, how the framework aids in the development of new algorithms or deepens our understanding of existing ones. Including more discussion on these points would greatly enhance the contribution of the work.

---

> ### Author Response · Authors · 2025-04-15
> **Preliminary response to reviewer j39o**
>
> Thank you sincerely for taking the time to read our manuscript and providing such prompt and constructive feedback, enabling us to begin improving the paper right away. We have only received your review so far, so here is our preliminary response. Once we receive the remaining reviews, we will provide a revised manuscript and another response to you.
>
> > Major concern 1: First, I am deeply concerned about whether MARL and game-theoretic algorithms can be accurately categorized under the umbrella of self-play. If they are to be considered as such, a clear and very general definition of self-play is necessary—something currently missing from the paper, and for which no consensus exists. In the case of classic self-play, it seems more accurate to view it as a special case of certain algorithms (e.g., PSRO), rather than considering PSRO as an instance of self-play. Even if a general definition were provided, I remain skeptical about whether such a definition is actually useful or necessary for the purposes of classification.
>
> We propose to add the following definition in the introduction section: "A learning process is called self‑play (SP) if at least one agent updates its policy by interacting with a distribution that contains copies or historical versions of itself (possibly mixed with other agents)".
>
> This definition goes beyond Vanilla SP (which interacts only with the latest version). It also captures PSRO [1] and its variants (each iteration optimizes against a meta‑strategy containing earlier policies), ongoing-training-based SP algorithms (training all the policies in the policy population together), and regret-minimization-based SP algorithms (computing or estimating and subsequently minimizing regret through SP).
>
> Under this definition, self‑play is central to landmark systems such as AlphaGo [2], AlphaStar [3], and Libratus [4], which all achieved super‑human performance. SP now drives emerging areas as well—for example, SPIN [5] for large language models (LLMs) and humanoid football [6] for robotics.
>
> Given its reach and impact, we believe self‑play merits a dedicated survey. Our goal is therefore twofold: (i) to summarise and organize the existing literature, and (ii) to offer unified insights to inspire future SP algorithms.
>
> [1] Lanctot, Marc, et al. "A unified game-theoretic approach to multiagent reinforcement learning." *Advances in neural information processing systems* 30 (2017).
>
> [2] Silver, David, et al. "Mastering the game of Go with deep neural networks and tree search." *nature* 529.7587 (2016): 484-489.
>
> [3] Vinyals, Oriol, et al. "Grandmaster level in StarCraft II using multi-agent reinforcement learning." *nature* 575.7782 (2019): 350-354.
>
> [4] Brown, Noam, and Tuomas Sandholm. "Superhuman AI for heads-up no-limit poker: Libratus beats top professionals." *Science* 359.6374 (2018): 418-424.
>
> [5] Chen, Zixiang, et al. "Self-Play Fine-Tuning Converts Weak Language Models to Strong Language Models." *International Conference on Machine Learning*. PMLR, 2024.
>
> [6] Liu, Siqi, et al. "From motor control to team play in simulated humanoid football." *Science Robotics* 7.69 (2022): eabo0235.
>
> > Major concern 2: Second, the self-play framework presented in the paper appears to subsume existing algorithms in a rather brute-force fashion. However, the paper does not clearly articulate the significance of this abstraction—specifically, how the framework aids in the development of new algorithms or deepens our understanding of existing ones. Including more discussion on these points would greatly enhance the contribution of the work.
>
> Thank you for this valuable suggestion. Section 3.6.1 (Categorization Rationale) explains how the framework helps us understand existing algorithms, but we acknowledge that its role in fostering new methods is not yet clear enough. To address this, we will add a dedicated subsection to Section 3.6 that presents how the framework can guide and inspire the design of novel self‑play algorithms.
>
> > Requested Changes: Besides the two concerns, there are some imprecise or incorrect descriptions of game theory preliminaries.
>
> Thank you for pointing out the imprecise descriptions in the game theory preliminaries and the other minor issues. Your detailed comments are extremely helpful. We will thoroughly revise the preliminaries section in the next version manuscript to ensure accuracy and clarity, and we will correct all minor errors.

---

> > ### Comment · Reviewer_j39o · 2025-05-07
> > **Further feedback**
> >
> > Thanks for the responses.
> >
> > You defined SP in "A learning process is called self‑play (SP) if at least one agent updates its policy by interacting with a distribution that contains copies or historical versions of itself (possibly mixed with other agents)." My first concern is whether this definition is too general. Can you provide some classes of methods that absolutely do NOT belong to SP based on your definition?
> >
> > My second concern is that if the game is not symmetric, then in many methods, a player interacts with other players' historical copies rather than its own. This does not align with your definition. But these methods should belong to your "SP" seemingly. Any thoughts on this?

---

> ### Author Response · Authors · 2025-05-10
> **Another response to reviewer j39o (Part 1)**
>
> Dear reviewer j39o,
>
> Thank you for your thorough evaluation of our survey and for the clear, constructive feedback you shared in your initial review. We have now completed a comprehensive revision of the manuscript that addresses every point you raised.
>
> > You defined SP in "A learning process is called self‑play (SP) if at least one agent updates its policy by interacting with a distribution that contains copies or historical versions of itself (possibly mixed with other agents)." My first concern is whether this definition is too general. Can you provide some classes of methods that absolutely do NOT belong to SP based on your definition? My second concern is that if the game is not symmetric, then in many methods, a player interacts with other players' historical copies rather than its own. This does not align with your definition. But these methods should belong to your "SP" seemingly. Any thoughts on this?
>
> Thank you for the "further feedback". We have refined the wording in **§1 Introduction** to make the scope of self-play (SP) precise in our revised manuscript:
>
> "Vanilla selfplay refers to a setting in which an agent continuously trains by competing against the most recent version of itself, as famously demonstrated for the symmetric zero-sum game of checkers (Samuel, 1959). Self-play, however, encompasses far more than this canonical example. More broadly, self-play refers to a class of learning methods where at least one agent **iteratively improves** its policy by interacting with **a distribution of evolving opponents**, which may include **historical or concurrent versions of itself or other agents**."
>
> For your first concern, self-play is characterised by two indispensable ingredients: an iterative round-by-round improvement loop and an evolving opponent population. Because both elements are required, many familiar paradigms remain outside the SP umbrella. For instance, single-agent reinforcement learning in a fixed environment—or, equivalently, training against an unchanging scripted bot—encounters no evolving adversary; ordinary MARL algorithms such as MAPPO never treat past versions as competitors; and offline RL or behavioural cloning relies on a frozen dataset rather than interactive rollouts.
>
> It is crucial to clarify how vanilla self-play, a special case of SP algorithms, differs from “ordinary” multi-agent RL. In vanilla self-play an agent first freezes the current policy, generates data by playing against that frozen copy, and only then replaces the copy with a newly updated version; the opponent distribution therefore changes across rounds, even though it contains only one element at any given instant. By contrast, in most standard MARL algorithms all agents update their parameters simultaneously while they are collecting data. Because the opponent distribution in that case has no well-defined “historical” component, it does not satisfy the round-by-round criterion in our definition of self-play.
>
> For your second concern, the revised sentence above deliberately permits the opponent distribution to contain “historical or concurrent versions of itself **or other agents**,” thereby subsuming such scenarios.
>
> > The self-play framework presented in the paper appears to subsume existing algorithms in a rather brute-force fashion. However, the paper does not clearly articulate the significance of this abstraction—specifically, how the framework aids in the development of new algorithms or deepens our understanding of existing ones. Including more discussion on these points would greatly enhance the contribution of the work.
>
> One motivation in proposing a unified self-play framework is to establish a common terminology and structure that spans the diverse range of self-play algorithms. In **§3.6.1 Categorization Rationale**, we explains how the framework helps us understand existing algorithms. A second motivation is to catalyse the development of new algorithms, and to that end we have added **§3.6.3 Implications for Future Self-Play Algorithm Design**. In addition to the well-studied ORACLE and MSS axes, this subsection introduces two further design levers within our proposed framework: (i) *prioritised retraining when E ≫ 1*, which recognizes that repeatedly updating every policy is wasteful once a single policy is reused across many encounters and therefore calls for principled scheduling; and (ii) *auxiliary information h(k) & knowledge transfer*, which treats the history kernel as an active carrier of domain knowledge that can be encoded to accelerate learning against new opponents and tasks. Together with ORACLE and MSS, these levers reveal a far richer combinatorial design space for future self-play research.

---

> > ### Author Response · Authors · 2025-05-10
> > **Another response to reviewer j39o (Part 2)**
> >
> > > In 2.2.1, “In a game with perfect information, only one player moves at a time. Each player comprehensively understands the current game state, the complete history of moves made, and the potential future outcomes.” First, “comprehensively understands” is vague. Second, this definition is a little bit informal. It would be better to define perfect information after defining the information sets, and then have a formal definition of perfect information in terms of information sets.
> >
> > Thank you for catching this. In the revision we now introduce information sets first and then give a formal definition of perfect (and imperfect) information (see the top of page 4 in the revised manuscript).
> >
> > > In 2.2.2, “The game is a zero-sum game, otherwise it is a general-sum game.” I think the key of general-sum is that the sum of the players’ payoffs is not constant. So it seems that the complementary part of general sum is constant sum, and zero sum is a special case of constant sum. It would be nice to clarify this relationship.
> >
> > We first define constant-sum games, in which the players’ payoffs always add to the same constant C; we then note that zero-sum games form the special case C=0; and finally we introduce general-sum games, where the payoff sum varies with the outcome. This sequence clarifies the relationship among the three categories (see the middle of page 3 in the revised manuscript).
> >
> > > The definition of symmetric games is incomplete. The missing part is that players should have the same utility function.
> >
> > We replaced the informal wording with a strict description (see the middle of page 3 in the revised manuscript): “If $\Pi_1=\cdots=\Pi_n$ and for any permutation $\tau$ and any strategy profile $(\pi_1, \dots, \pi_n)$, $u_i(\pi_1,\cdots,\pi_n) = u_{\tau(i)}(\pi_{\tau(1)},\cdots,\pi_{\tau(n)})$, the game is a symmetric game.”  This permutation invariance guarantees that all players effectively have the same utility function, whereas merely assigning identical utility functions does not, by itself, ensure permutation invariance.
> >
> > > For the last line on page 3, little pi was defined as a pure strategy at the beginning of 2.2.2, while here pi is defined as a behavior strategy, which seems inconsistent. I suggest to follow the pure-strategy and mixed-strategy definitions at the beginning of 2.2.2 and define behavior strategy separately. Then you can describe the equivalence between mixed strategy and behavior strategy under certain conditions. In addition, a typo “A” should be “a”.
> >
> > We have eliminated the inconsistency by reserving $\pi$ for pure strategies and introducing $\beta$ for behaviour strategies. After formally defining perfect recall, we invoke Kuhn’s theorem and state that, in any finite extensive-form game with perfect recall, every mixed strategy is outcome-equivalent to a behaviour strategy, and vice versa (see the top of page 4 in the revised manuscript). We fix the typo as well.
> >
> > >In the next sentence “A subgame of an extensive-form game is a portion of the game that starts from a single initial node, includes all successors of any node within the subgame, and contains all nodes in the same information set as any node in the subgame.” This definition seems to be circular since a subgame is defined in terms of subgame.
> >
> > We have removed the circular wording and now give a non-recursive definition (see the top of page 4 in the revised manuscript):
> >
> > "A subgame of an extensive-form game is any subset of the game tree that (i) has exactly one initial decision node, (ii) whenever it contains a node it also contains all of its descendants, and (iii) whenever it contains a node it contains every other node in that node’s information set."
> >
> > > In the Prisoner’s dilemma examples for Fig.2, it is less interesting to have or compare the simultaneous-move version with the sequential move version since Defect (C) is a dominant strategy. Maybe there are more interesting examples to illustrate the difference between static games and dynamic games.
> >
> > Thank you for the suggestion. In **§2.2.5 Static Game and Dynamic Game**, we now use the *market entry* scenario in place of the Prisoner’s Dilemma. In the static (simultaneous‑move) version the game admits two pure‑strategy NE, one of which caused by a non‑credible threat; when we rewrite the same situation as a dynamic game and analyse it with subgame‑perfect Nash equilibrium (SPNE), only one NE survives. We believe this substitution illustrates the difference between static and dynamic reasoning more interestingly.

---

> > > ### Author Response · Authors · 2025-05-10
> > > **Another response to reviewer j39o (Part 3)**
> > >
> > > > For the definition of meta-games, I think meta-games are not limited to be a normal-form game. They can take any form (e.g., EFG) as long as they provide a certain abstraction of the original game.
> > >
> > > We have revised the description of meta-games (see the middle of page 4 in the revised manuscript) to clarify that they are not confined to normal-form representations. A meta-game may adopt any game-theoretic format—including extensive form—so long as it abstracts the strategic interactions among the policies in the population.
> > >
> > > > In Algorithm 3, line 3 is hand-waving.
> > >
> > > Thank you for pointing this out. We have updated line 3 of Algorithm 3 to replace the previous hand-waving description with a precise step and added the appropriate citation. Although the underlying reward transformation is $r^i(\pi^i\pi^{-i}a^ia^{-1}) =r^i(a^ia^{-1}) -\eta\mathrm{log}\left(\frac{\pi^i(a^i)}{\pi_{\mathrm{reg}}^i(a^i)}\right) +\eta\mathrm{log}\left(\frac{\pi^{-i}(a^{-i})}{\pi_{\mathrm{reg}}^{-i}(a^{-i})}\right)$. This detail is peripheral to our survey. Accordingly, the algorithm box now reads: "Transform the reward by adding a policy-dependent reward according to $\pi_{\mathrm{reg}} $ [1]"
> > >
> > > This keeps the pseudocode clear and concise while directing readers to the full formula in the cited work.
> > >
> > > [1] Perolat, Julien, et al. "From poincaré recurrence to convergence in imperfect information games: Finding equilibrium via regularization." *International Conference on Machine Learning*. PMLR, 2021.
> > >
> > > > For Eq.6, I would say it is just fictitious play applied to meta games rather than FSP.
> > >
> > > Thank you for the clarification. We’ve updated **§3.2.3 Fictitious Play** to make it explicit that Eq. 6 is simply the standard fictitious-play update applied to the meta-game’s payoff structure, rather than FSP.
> > >
> > > > Typos: activa -> active, MSSes -> MSSs
> > >
> > > Thank you for catching these typos. We have corrected “activa” to “active” and updated “MSSes” to “MSSs” in the revised manuscript.

---

### Review · Reviewer_Eyth · 2025-04-16

**Summary Of Contributions:**

This paper provides a comprehensive survey of self-play algorithms by first proposing a unified self-play algorithm and then providing a detailed discussion on how to integrate different categories of self-play algorithms into this unified framework, including traditional self-play, PSRO, ongoing-training-based, and regret-minimization-based algorithms. The authors also provided a discussion on the advantages and limitations of the self-play algorithms, which could motivate further research.

**Audience:**

Yes

**Broader Impact Concerns:**

There is no ethical concern for this paper.

**Claims And Evidence:**

Yes

**Requested Changes:**

See **Strengths And Weaknesses**. Some more discussions on the motivation of a unified framework, and multi-player scenarios could be helpful. Also, the references should be checked and updated.

**Strengths And Weaknesses:**

This paper provides a comprehensive survey of self-play algorithms, which provides good insights into their current position in a broader research area. I think this is a nice survey, where the authors propose a unified framework to characterize different categories of self-play algorithms, including traditional self-play, PSRO, ongoing-training-based, and regret-minimization-based algorithms. Importantly, the authors provide how these different algorithms can be integrated into the unified framework, which could clearly show the relationships and differences between these algorithms and could motivate future advancements. Besides, the authors also discuss the advantages and limitations of self-play algorithms, which could provide insights on how to further improve the algorithms from different aspects.

Some minor comments:

1. This paper primarily focuses on two-player zero-sum games. Maybe some discussion on multi-player games could be valuable, as in real-world applications, there are often more than two players.

2. Although a unified framework could be desired, maybe some discussion on the motivation could be helpful, e.g., how could such a unified framework benefit real-world applications?

3. When discussing the PSRO and ongoing-training-based series of algorithms, some relevant references have not been mentioned: [1] transferred knowledge from previous iterations to the current iteration to accelerate the convergence, [2,3] used pre-trained policies to accelerate the learning process of PSRO.

4. The format of the references should be checked and improved. Many references are the arXiv version, but many of them have been published in conferences or journals. SPAG was published in NeurIPS 2024, not NeurIPS 2025.

----

[1] Max Olan Smith et al. Iterative Empirical Game Solving via Single Policy Best Response. ICLR 2021.

[2] Pengdeng Li et al. Grasper: A Generalist Pursuer for Pursuit-Evasion Problems. AAMAS 2024.

[3] Shuxin Li et al. Solving Large-Scale Pursuit-Evasion Games Using Pre-Trained Strategies. AAAI 2023.

---

> ### Author Response · Authors · 2025-05-10
> **Response to reviewer Eyth (Part 1)**
>
> Dear reviewer Eyth,
>
> Thank you for the time and care you devoted to evaluating our survey. We have carefully considered every concern you raised and provide point-by-point responses below.
>
> > This paper primarily focuses on two-player zero-sum games. Maybe some discussion on multi-player games could be valuable, as in real-world applications, there are often more than two players.
>
> Thank you for pointing this out. We agree that many real-world problems involve more than two decision-makers. Our manuscript emphasises two-player zero-sum games mainly because their well-understood structure lets us present definitions, examples, and theoretical results with minimal overhead, and because the majority of historical self-play research has used this setting as a proving ground. Nevertheless, the framework we propose is *not* restricted to two-player zero-sum games. In **§3.1 Framework Definition**, we note that, for an n-player game, the interaction tensor can be written as  $\Sigma\in\mathbb{R}^{K\times K\times\cdots\times K}$ and the evaluation operator EVAL($\Pi$) naturally returns a performance tensor $P\in\mathbb{R}^{K\times K\times\cdots\times K}$. Crucially, neither $\Sigma$ nor $P$ must satisfy a zero-sum constraint.
>
> Multi-player algorithms are covered explicitly in **§3.3.4 PSRO Variants**, where we review $\alpha$-PSRO [1] and JPSRO [2]. They target at solving multi-player games. In section **§4.2 Card Games** and **§4.3 Video Games**, we also discuss multi-agent scenarios, such as Pluribus [3] for six-player no-limit Texas Hold’em, OpenAI Five [4] and MOBA AI [5] for 5v5 MOBA games, and Tizero [6] and FXP [7] for 11v11 GRF.
>
> [1] Muller, P., et al. "A Generalized Training Approach for Multiagent Learning." *ICLR*. ICLR, 2020.
>
> [2] Marris, Luke, et al. "Multi-agent training beyond zero-sum with correlated equilibrium meta-solvers." *International Conference on Machine Learning*. PMLR, 2021.
>
> [3] Brown, Noam, and Tuomas Sandholm. "Superhuman AI for multiplayer poker." *Science* 365.6456 (2019): 885-890.
>
> [4] Berner, Christopher, et al. "Dota 2 with large scale deep reinforcement learning." *arXiv preprint arXiv:1912.06680* (2019).
>
> [5] Ye, Deheng, et al. "Towards playing full moba games with deep reinforcement learning." *Advances in Neural Information Processing Systems* 33 (2020): 621-632.
>
> [6] Lin, Fanqi, et al. "TiZero: Mastering Multi-Agent Football with Curriculum Learning and Self-Play." *Proceedings of the 2023 International Conference on Autonomous Agents and Multiagent Systems*. 2023.
>
> [7] Xu, Zelai, et al. "Fictitious Cross-Play: Learning Global Nash Equilibrium in Mixed Cooperative-Competitive Games." *Proceedings of the 2023 International Conference on Autonomous Agents and Multiagent Systems*. 2023.
>
> > Although a unified framework could be desired, maybe some discussion on the motivation could be helpful, e.g., how could such a unified framework benefit real-world applications?
>
> To clarify the practical motivation for a unified self-play framework, we have inserted a new subsection, we have inserted a new section **§3.6.3 Implications for Future Self-Play Algorithm Design** in **§3.6 Reassessment of the Framework**. In addition to the well-studied ORACLE and MSS axes, this subsection introduces two further design levers within our proposed framework: (i) *prioritised retraining when E ≫ 1*, which recognizes that repeatedly updating every policy is wasteful once a single policy is reused across many encounters and therefore calls for principled scheduling; and (ii) *auxiliary information h(k) & knowledge transfer*, which treats the history kernel as an active carrier of domain knowledge that can be encoded to accelerate learning against new opponents and tasks. Viewed alongside ORACLE and MSS, these levers open a combinatorial design space that can be tuned to real-world constraints. Complementing this analysis, **§5.5 Real-World Applications** now discusses application bottlenecks such as the high cost and safety risk of trial-and-error in physical systems, the reliance on simulators, and the resulting sim-to-real gap, and shows how the unified framework highlights where sample-efficient, online-capable self-play algorithms are urgently needed.

---

> ### Author Response · Authors · 2025-05-10
> **Response to reviewer Eyth (Part 2)**
>
> > When discussing the PSRO and ongoing-training-based series of algorithms, some relevant references have not been mentioned: [1] transferred knowledge from previous iterations to the current iteration to accelerate the convergence, [2,3] used pre-trained policies to accelerate the learning process of PSRO.
>
> When discussing the PSRO series of algorithms, we initially overlooked these pertinent studies. We have now integrated them into **§3.3.4 PSRO Variants**. Smith et al. [1] proposes Mixed-Oracles and MixedOpponents, which modify how new policies are added to the empirical game by training responses to a single opponent policy, thereby reducing the cost of simulation. Li et al. [2,3] enhance the scalability of PSRO for large-scale pursuit-evasion games by incorporating pre-training and fine-tuning paradigms into the policy learning process.
>
> [1] Max Olan Smith et al. Iterative Empirical Game Solving via Single Policy Best Response. ICLR 2021.
>
> [2] Pengdeng Li et al. Grasper: A Generalist Pursuer for Pursuit-Evasion Problems. AAMAS 2024.
>
> [3] Shuxin Li et al. Solving Large-Scale Pursuit-Evasion Games Using Pre-Trained Strategies. AAAI 2023.
>
> > The format of the references should be checked and improved. Many references are the arXiv version, but many of them have been published in conferences or journals. SPAG was published in NeurIPS 2024, not NeurIPS 2025.
>
> Thank you for mentioning these issues. We have thoroughly audited the entire bibliograph replacing arXiv citations with their corresponding conference or journal versions wherever availabl and have corrected the SPAG entry.

---

### Review · Reviewer_Jb4T · 2025-04-25

**Summary Of Contributions:**

The authors present a survey of RL and RL-adjecent self-play methods for multi-agent games. They also provide some example domains where the algorithms have been successfully applied.. The article places the algorithms within a iterative, population-based framework.

**Audience:**

Yes

**Claims And Evidence:**

Yes

**Requested Changes:**

3.6.1 “making it particularly suitable for repeated games. For example, Texas Hold’em is a classic repeated game where players adjust their strategies based on past interactions and use tactics involving deception and bluffing.”   and the similar statement in 4.2
I don’t follow this argument. All the algorithms are trying to generate strong (for some metric) policies for multi-agent games. The regret-minimizing algorithms are mostly designed around approximating a NE in two-player zero-sum games, which is also one of the goals of PSRO. Minimizing regret is a technique for online play, which could describe the repeated game scenario, but the CFR-based methods in section 3.5 are not using it in this fashion. That is, they are not actually being used in an online fashion to handle repeated play, but in an offline fashion to generate a policy for the stage game. How is one approximate NE algorithm more suitable for repeated play than another?

—
3 and 4.1.1 It seems like there is another possible set of self-play algorithms, intended to deal with cooperative games. E.g., Cui et. al “K-level Reasoning for Zero-Shot Coordination in Hanabi” and earlier papers where Jakob Foerster has been involved
Or more broadly, algorithms for communication and coordination games. While I’m not particularly familiar with the area, some sort of modified self-play seems to my recollection to be fairly common.

RL / MCTS self-play in the style of AlphaGo / AlphaZero is mentioned here – why aren’t these methods also in Section 3?
If the authors’ response would be that RL+MCTS algorithms would just be an extreme variant of vanilla self-play, that makes that section 3.2.2 category so broad that it should be expanded to mention these variants.

4.3 Overcooked seems like a fairly common MARL testbed that is missing here.

—
3.5.1 Why does the regret-minimizing series have such detailed explanations, where the others do not?  In particular, why describe the underlying theory involving immediate regret and bounds, rather than describing the algorithm as minimizing counterfactual regrets at information sets, and note that the cited paper shows that is enough to minimize regret?

Contrast this to section 3.3.4, which has one or two sentence descriptions of many algorithms. What parts of the framework-based description changes? Do they actually all fit? For example, the incremental tree of XDO goes into h?


— smaller issues —
2.2.2  “specifies a particular and deterministic action”
The phrasing using “particular” was unclear to me.  Maybe “specifies a deterministic action choice for each point in the game” ?

2.2.2 This section introduces zero sum and general sum. Also a place to mention cooperative?

2.2.2 “where u_i: Pi -> R, is a utility function that assigns a real-valued payoff to each player i.”
Expected payoff? Given stochastic events in the underlying game being described, there is a distribution of trajectories seen when following a strategy profile, even if all strategies in the profile are pure.

2.2.2 “can be directly depicted in a matrix”  two player zero-sum games?  Two player general sum games are more naturally represented with 2 matrices or a tensor. With more than two players, the natural representation seems to be a >2D tensor.

2.2.2 “if player i has perfect recall, it means that player i remembers which action”
They remember all their actions, and all previously observed state.

2.2.2 My understanding is that sequential Prisoner’s Dilemma usually describes the repeated game.
Also, “with knowledge of the first player’s action” reads to me as describing player 2 knowing what player 1’s action was, as opposed to player 2 only knowing player 1 has made an action. This is reinforced by the phrasing “Another variant” and the extensive form tree where both histories “C” and “L” are not shown as being in a single information set. I suggest re-wording with something like “Another way to describe the game is” and “with the knowledge that they are second to act”, and modifying the figure to show the player 2 information set describing player 2’s lack of knowledge of player 1’s choice.

2.2.3 “For the sake of simplicity, we restrict our focus to two-player zero-sum symmetric games”
Is this comment only intended to apply to this section? If so, I suggest re-wording to make that clear.   (And if it is intended for the whole article, I would suggest adding this to the abstract)

3 “All players share a policy population with a fixed maximum size… After training, the new policy is added to the  population.”
suggest “added to the population, possibly replacing an existing policy”.

3.1 “N denotes the policy population size, not the number of players”
Use a different letter: K for capacity? Or something other than N for number of players, if the authors are concerned about catching all uses throughout the rest of the paper? The paper is introducing the framework and its notation, and you as the authors have the freedom to not re-use a letter in a way that immediately needs clarification. Don’t introduce notation, then switch.
There is a similar issue with the notational switch from pi_i being a player i strategy to pi_i being the i’th strategy in a finite set.

3.2.2 How does the replay buffer fit into the framework?

3.3.2 Standard double oracle method, as in the McMahan reference, adds a pure best response. This (population based?) variant which is adding a mixed best response strategy should have a reference.

3.5 It seems worth noting that all of these methods are most strongly theoretically justified in two-player zero-sum environments.

3.5.1 “More specifically, h(i) = h(i−1)” and “h(i) represents the specific elements that regret-minimization-based self-play algorithms need to store”
If h(i) = h(i-1), how is it storing an accumulated set of regrets?

3.5.1 “Strategies  will converge to an approximate NE”
Convergence is to the set of NE, with any finite stopping point being an approximate NE

4.3.2 “hero pool to 40 heroes, exponentially increasing the possible combinations of lineups”
(lots of text, but a small issue) exponential describes a growth pattern, not a magnitude of increase. Even if the number of possible combinations was actually increasing exponentially, it is not strictly correct to say that the increase going from A to B is exponential. Unless I misunderstand what is meant by combinations of lineups, the number of 5 player lineups is N choose 5, an order 5 polynomial.
I suggest just using “substantially increasing”, and also listing the size of the hero pool used for Five in the paragraph describing it.

5.1 “higher levels of equilibrium, such as CE”
suggest “alternative solution concepts, such as CE”.  Higher level seems (i) ill defined, and (ii) backwards with respect to CE, at least  to me personally, as I would have expected “higher level” to correspond to NE refinements (stricter conditions and smaller solution set, like the quasi-perfect equilibrium).

“they lack formal game theory proofs behind their effectiveness”
This might need to be carefully softened, or caveats added. It’s not a particularly satisfying argument for finite computational bounds, but almost any UCT-like variant will eventually do the right thing in two player, competitive, perfect information games when the entire game tree has been built (possibly with high probability, depending on the variant)

5.4 “A representative work is Cicero (, FAIR)”
broken citation

**Strengths And Weaknesses:**

Strengths
This article considers a broader range of self-play algorithms than some existing surveys.

Weaknesses
I’m not sure I understand what the framework is providing within the survey.
There does seem to be a potential cost. The description of a number of algorithms so they fit into the framework seems like a more complicated, less natural way to describe them.  (I may be biased towards their classical descriptions as it’s what I’m familiar with)
There seems to be a partial commitment to using the framework: rather than describing an algorithm with the framework’s parts and then potentially clarifying, algorithms seem to have a general description and then some argumentation that it fits within the framework.
The framework is broad enough to contain all of the algorithms in section 3 and 4. Are there conclusions we can now draw by looking at the algorithms placed within that framework?

---

> ### Author Response · Authors · 2025-05-10
> **Response to reviewer Jb4T (Part 1)**
>
> Dear reviewer Jb4T,
>
> Thank you for the time and care you devoted to evaluating our survey. We have carefully considered every concern you raised and provide point-by-point responses below.
>
> >Weaknesses I’m not sure I understand what the framework is providing within the survey. There does seem to be a potential cost. The description of a number of algorithms so they fit into the framework seems like a more complicated, less natural way to describe them. (I may be biased towards their classical descriptions as it’s what I’m familiar with) There seems to be a partial commitment to using the framework: rather than describing an algorithm with the framework’s parts and then potentially clarifying, algorithms seem to have a general description and then some argumentation that it fits within the framework. The framework is broad enough to contain all of the algorithms in section 3 and 4. Are there conclusions we can now draw by looking at the algorithms placed within that framework?
>
> The first motivation in proposing a unified self-play framework is to establish a common terminology and structure that spans the diverse range of self-play algorithms. By formulating a single framework that can describe traditional self-play, PSRO-based methods, ongoing training approaches, and regret-minimization techniques, we aim to unify the language used to discuss these algorithms. This unified formalism makes it easier to compare methods side by side – algorithms that were originally described in disparate terms can now be expressed within the same set of concepts. In **§3.6.1 Categorization Rationale**, we demonstrate how the framework clarifies existing algorithms, making their shared foundations and distinguishing features explicit.
>
> The second motivation is to catalyse the development of new algorithms, and to that end we have added **§3.6.3 Implications for Future Self-Play Algorithm Design**. In addition to the well-studied ORACLE and MSS axes, this subsection introduces two further design levers within our proposed framework: (i) *prioritised retraining when E ≫ 1*, which recognizes that repeatedly updating every policy is wasteful once a single policy is reused across many encounters and therefore calls for principled scheduling; and (ii) *auxiliary information h(k) & knowledge transfer*, which treats the history kernel as an active carrier of domain knowledge that can be encoded to accelerate learning against new opponents and tasks. Together with ORACLE and MSS, these levers reveal a far richer combinatorial design space for future self-play research.
>
> We keep Sections 3 and 4 separate for clarity. Section 3 isolates the self-play core of each algorithm, stripping away domain-specific components so that the reader can see exactly how different methods instantiate the framework. Section 4 then shows how that core is paired with additional machinery—such as MCTS for Go—to solve specific scenarios. Presenting the material in this order lets us first build a clean theoretical self-play foundation and then demonstrate, in a modular way, how self-play combines with other techniques to achieve practical success.
>
> > 3.6.1 “making it particularly suitable for repeated games. For example, Texas Hold’em is a classic repeated game where players adjust their strategies based on past interactions and use tactics involving deception and bluffing.” and the similar statement in 4.2 I don’t follow this argument. All the algorithms are trying to generate strong (for some metric) policies for multi-agent games. The regret-minimizing algorithms are mostly designed around approximating a NE in two-player zero-sum games, which is also one of the goals of PSRO. Minimizing regret is a technique for online play, which could describe the repeated game scenario, but the CFR-based methods in section 3.5 are not using it in this fashion. That is, they are not actually being used in an online fashion to handle repeated play, but in an offline fashion to generate a policy for the stage game. How is one approximate NE algorithm more suitable for repeated play than another?
>
> We have removed the earlier claim that regret-minimisation methods are “particularly suitable for repeated games.” In both **§3.6.1** and **§4.2** we now state the point more precisely: CFR and its variants are designed for *imperfect-information* games and excel at computing approximate Nash equilibria when some information is hidden, as in Texas Hold’em. The revised text no longer implies that CFR is uniquely suited to repeated play; instead, it highlights CFR’s strength in dealing with hidden information relative to other equilibrium-finding algorithms.

---

> ### Author Response · Authors · 2025-05-10
> **Response to reviewer Jb4T (Part 2)**
>
> > — 3 and 4.1.1 It seems like there is another possible set of self-play algorithms, intended to deal with cooperative games. E.g., Cui et. al “K-level Reasoning for Zero-Shot Coordination in Hanabi” and earlier papers where Jakob Foerster has been involved Or more broadly, algorithms for communication and coordination games. While I’m not particularly familiar with the area, some sort of modified self-play seems to my recollection to be fairly common.
>
> Thank you for highlighting work on cooperative-game settings such as zero-shot coordination in Hanabi. Although these methods also train agents against past or concurrent teammates, their objective is to maximise a shared reward, placing them squarely in the cooperative MARL literature. Our survey, by contrast, analyzes self-play as a tool for non-cooperative games, we deliberately confined the scope to methods whose evaluation hinges on strategic opposition rather than team performance. To make this boundary clear, we now state it explicitly in both the abstract and the Introduction and cite representative coordination-game papers [1, 2] so that readers interested in cooperative applications can consult the relevant work.  As the Introduction now states:
>
> "It is worth noting that although some MARL studies have applied the concept of self-play in cooperative tasks such as Hanabi (Cui et al., 2021) and Overcooked (Strouse et al., 2021), our survey focuses on its original and most prevalent application—non-cooperative tasks."
>
> [1] Cui, Brandon, et al. "K-level reasoning for zero-shot coordination in hanabi." *Advances in Neural Information Processing Systems* 34 (2021): 8215-8228.
>
> [2] Strouse, D. J., et al. "Collaborating with humans without human data." *Advances in Neural Information Processing Systems* 34 (2021): 14502-14515.
>
> > RL / MCTS self-play in the style of AlphaGo / AlphaZero is mentioned here – why aren’t these methods also in Section 3? If the authors’ response would be that RL+MCTS algorithms would just be an extreme variant of vanilla self-play, that makes that section 3.2.2 category so broad that it should be expanded to mention these variants.
>
> We keep Sections 3 and 4 separate to maintain clarity. Section 3 isolates the self-play core of each algorithm, stripping away domain-specific elements so readers can see exactly how each method instantiates the framework. Techniques such as AlphaGo/AlphaZero combine that core with additional machinery—Monte-Carlo Tree Search, domain heuristics—so they are treated in Section 4, which is devoted to such integrations. To bridge the two sections, we have added a short remark in **§3.2.2 Vanilla Self-Play** noting that vanilla self-play is often paired with MCTS. Moreover, in the revised manuscript, **§4.4.1 Go** now offers an expanded, more detailed account of the AlphaGo lineage, illustrating step by step how MCTS is layered on top of the vanilla self-play loop.
>
> > 4.3 Overcooked seems like a fairly common MARL testbed that is missing here.
>
> Overcooked is indeed a popular cooperative MARL benchmark. Because our survey focuses on self-play for non-cooperative games, we do not include Overcooked in the main taxonomy. To acknowledge the parallel literature, however, we have added a citation to a representative Overcooked study [1] in the Introduction, where we briefly note that such fully co-operative applications lie outside the scope of our review.
>
> [1] Strouse, D. J., et al. "Collaborating with humans without human data." *Advances in Neural Information Processing Systems* 34 (2021): 14502-14515.
>
> > — 3.5.1 Why does the regret-minimizing series have such detailed explanations, where the others do not? In particular, why describe the underlying theory involving immediate regret and bounds, rather than describing the algorithm as minimizing counterfactual regrets at information sets, and note that the cited paper shows that is enough to minimize regret?
>
> We have rewritten **§3.5.2 Vanilla CFR** to remove the heavy theoretical derivations and focus instead on the core idea. The new text explains that CFR stores counterfactual regret at each information set. The global regret-minimisation task can be decomposed into many local sub-problems. It then states the central theoretical guarantee: in two-player zero-sum games, the averaged strategy over time converges to an NE.

---

> ### Author Response · Authors · 2025-05-10
> **Response to reviewer Jb4T (Part 3)**
>
> > Contrast this to section 3.3.4, which has one or two sentence descriptions of many algorithms. What parts of the framework-based description changes? Do they actually all fit? For example, the incremental tree of XDO goes into h?
>
> Although the original PSRO algorithm maps cleanly onto our framework, some of its extensions do not fit as neatly; nevertheless, they introduce valuable ideas and warrant discussion. We now make this point explicit at the beginning of **§3.3.4 PSRO Variants**. The revised subsection organizes the variants around three themes: (i) techniques that accelerate convergence, (ii) methods that improve computational tractability, and (iii) approaches that encourage policy diversity.
>
> > — smaller issues — 2.2.2 “specifies a particular and deterministic action” The phrasing using “particular” was unclear to me. Maybe “specifies a deterministic action choice for each point in the game” ?
>
> Yes. We now write: “A pure strategy specifies a deterministic action choice for each point in the game.” (see the middle of page 3 in the revised manuscript). It is worth noting that we have moved the perfect- and imperfect-information definitions into the introductions of the normal-form and extensive-form sections, and have renumbered the former §2.2.2 as §2.2.1 to reflect these changes.
>
> > 2.2.2 This section introduces zero sum and general sum. Also a place to mention cooperative?
>
> Our survey focuses exclusively on non-cooperative games, so we do not cover cooperative settings in §2.2.1 To improve clarity, we have reordered that section to introduce constant-sum games first, then zero-sum as the special case C=0, and finally general-sum games, making the relationships among these categories more explicit.
>
> > 2.2.2 “where u_i: Pi -> R, is a utility function that assigns a real-valued payoff to each player i.” Expected payoff? Given stochastic events in the underlying game being described, there is a distribution of trajectories seen when following a strategy profile, even if all strategies in the profile are pure.
>
> Thanks for the suggestion. We have updated the wording in §2.2.1 to clarify that each player’s utility represents their “expected payoff” under a given strategy profile.
>
> > 2.2.2 “can be directly depicted in a matrix” two player zero-sum games? Two player general sum games are more naturally represented with 2 matrices or a tensor. With more than two players, the natural representation seems to be a >2D tensor.
>
> In the revised manuscript (see §2.2.1), we now describe a general n-player normal-form game using a payoff tensor $T\in\mathbb{R}^{|\Pi_1|\times\cdots\times|\Pi_n|\times n}$, where $|\Pi_i|$ is the size of player $i$'s strategy space,  and the final dimension indexes each player’s payoff. We then explain that in the special case of a two-player zero-sum symmetric game, this tensor collapses to a single evaluation matrix $M_\Pi\in\mathbb{R}^{|\Pi|\times|\Pi|}$.
>
> > 2.2.2 “if player i has perfect recall, it means that player i remembers which action” They remember all their actions, and all previously observed state.
>
> We have revised the definition of perfect recall to:
>
> “A game satisfies perfect recall if each player i remembers the entire sequence of his past actions and all of the information he had at each of his previous decision points.”

---

> ### Author Response · Authors · 2025-05-10
> **Response to reviewer Jb4T (Part 4)**
>
> > 2.2.3 “For the sake of simplicity, we restrict our focus to two-player zero-sum symmetric games” Is this comment only intended to apply to this section? If so, I suggest re-wording to make that clear. (And if it is intended for the whole article, I would suggest adding this to the abstract)
>
> This restriction applies only to §2.2.2. At the start of that section we now write:
>
> "To provide a clearer introduction to transitive and non-transitive games, in this section, we confine our analysis to two-player zero-sum symmetric games."
>
> This makes it clear that the simplification is local to this section and not a limitation on the entire paper.
>
> > 3 “All players share a policy population with a fixed maximum size… After training, the new policy is added to the population.” suggest “added to the population, possibly replacing an existing policy”.
>
> Thank you for the suggestion. We have updated the text to clarify this point. It now reads:
>
> “After training, the new policy is added to the population, possibly replacing an existing policy.”
>
> > 3.1 “N denotes the policy population size, not the number of players” Use a different letter: K for capacity? Or something other than N for number of players, if the authors are concerned about catching all uses throughout the rest of the paper? The paper is introducing the framework and its notation, and you as the authors have the freedom to not re-use a letter in a way that immediately needs clarification. Don’t introduce notation, then switch. There is a similar issue with the notational switch from pi_i being a player i strategy to pi_i being the i’th strategy in a finite set.
>
> We sincerely appreciate the reviewer’s meticulous attention to notational clarity, which is critical for ensuring the framework’s interpretability. The variable N, originally denoting both policy population size and player count, has been replaced with K to exclusively represent capacity constraints. Regarding the subscript ambiguity in strategy notation, we have introduced a distinct bracketed index format  $\pi_{[k]}$ to unambiguously denote the k-th policy within the population.  These refinements have been applied consistently across the manuscript, including in the main text, all figures, and the pseudocode.
>
> > 3.2.2 How does the replay buffer fit into the framework?
>
> We’ve updated Algorithm 2 to make the replay buffer an explicit part of the ORACLE component. Now, at each iteration the ORACLE subroutine collects new self-play trajectories into the buffer and then updates the policy by sampling experiences from that buffer.
>
> > 3.3.2 Standard double oracle method, as in the McMahan reference, adds a pure best response. This (population based?) variant which is adding a mixed best response strategy should have a reference.
>
> We discovered that an extra iteration in our implementation was inadvertently producing a mixed‐strategy best response in the final round. We have fixed this by capping the policy population size at K=3 and strictly following the double‐oracle schedule from McMahan et al. (2003). With this correction, our method now adds only pure best responses at each step, fully matching the standard double‐oracle procedure.
>
> > 3.5 It seems worth noting that all of these methods are most strongly theoretically justified in two-player zero-sum environments.
>
> Thank you for the suggestion. We have revised §3.5 to explicitly note that most of these regret-minimization methods enjoy their strong theoretical guarantees in two-player zero-sum settings, where they are proven to converge to NE.
>
> > 3.5.1 “More specifically, h(i) = h(i−1)” and “h(i) represents the specific elements that regret-minimization-based self-play algorithms need to store” If h(i) = h(i-1), how is it storing an accumulated set of regrets?
>
> In the revised manuscript §3.5.1, we further clarify how $h$ is updated. At each iteration k, we begin by $h(k)$ initialized from $h(k −1) $ and $π_{h [k]}$ initialized from $π_{h [k-1]}$. In each new iteration, regret-minimization information is continuously added to $h(k)$ (Line 4 in Algo. 4). As a result, $h$ accumulates information across all iterations until the end of training.
>
> > 3.5.1 “Strategies will converge to an approximate NE” Convergence is to the set of NE, with any finite stopping point being an approximate NE
>
> Thank you for the precision. In §3.5.2 we now clarify that the algorithm is guaranteed to converge to the set of NE, and any finite stopping point yields an approximate NE.

---

> ### Author Response · Authors · 2025-05-10
> **Response to reviewer Jb4T (Part 5)**
>
> > 5.1 “higher levels of equilibrium, such as CE” suggest “alternative solution concepts, such as CE”. Higher level seems (i) ill defined, and (ii) backwards with respect to CE, at least to me personally, as I would have expected “higher level” to correspond to NE refinements (stricter conditions and smaller solution set, like the quasi-perfect equilibrium).
>
> Thank you for the suggestion. In §5.1 we have replaced the phrase “higher levels of equilibrium, such as CE” with “alternative solution concepts, such as CE,” avoiding the ambiguous “higher level” terminology.
>
> > “they lack formal game theory proofs behind their effectiveness” This might need to be carefully softened, or caveats added. It’s not a particularly satisfying argument for finite computational bounds, but almost any UCT-like variant will eventually do the right thing in two player, competitive, perfect information games when the entire game tree has been built (possibly with high probability, depending on the variant)
>
> Thank you for the note. In §5.1 we have softened our wording and added important caveats to avoid the blanket “lack formal proofs” claim and acknowledges both their convergence properties and the limitations of finite computational budgets.
>
> > 5.4 “A representative work is Cicero (, FAIR)” broken citation
>
> Thank you for catching the broken citation in §5.4. We’ve corrected it in the revised manuscript.

---

> ### Comment · Reviewer_Jb4T · 2025-05-20
> **Response to author response**
>
> Thank you to the authors for the response, and the corrections of previous issues.
>
>
> Part of my response was poorly worded, and it was read as me suggesting that sections 3 and 4 should be merged. I only grouped together 3 and 4 to talk about all the algorithms and all the implementations, all at once. My intended point was that it’s clear that the framework is broad enough to cover many algorithms, but that it isn’t clear what that classification offers. That is, I could have an even broader class based only on being iterative algorithms with an initialization and final selection, but this isn’t going to offer much insight into the members because it’s so broad that every possible algorithm fits. In the case of the proposed framework, the new section 3.6.3 is a step towards showing utility for the framework (i.e. the restrictions on what fits within the framework may tell us something about that set of algorithms). However, I’m not sure that hypothetical future research directions make a strong enough case on their own.
>
> The restriction to non-cooperative (but not strictly competitive) ends up with the result where population-based self-play algorithms using shared knowledge for cooperative games are ruled out as too dissimilar to things like PSRO, while R-NAD and CFR are included. There is a claim that the formalism “makes it easier to compare methods side by side”. I am aware that this is a subjective judgment, but I do disagree, largely because I think the formalism is not a natural fit for all the algorithms. What should I as a reader take from noting that CFR is an iterative method with no population, where I can generate the next strategy from some auxiliary information?
>
>
> ----------------------------------------
> There were substantial changes to the text, so there are some suggestions addressing some potential issues in the new text.
>
> “further exacerbating the non-stationarity problem”
> claim needs justification? cooperative self-play also has problems with the strategies being non-stationary.
>
> “focuses on its original and most prevalent application”
> Is it really the most prevalent? Applications to team coordination: drones, warehouse pathing, and distributed planning seem common, and could arguably be more commonly applied.
>
> “with a particular focus on non-cooperative settings”
> “within non-cooperative settings”   it’s a restriction, not just a focus
>
> “The normal-form representation is appropriate here, as it captures the strategic interaction under imperfect information.”
> “Games of this nature are called dynamic games. The extensive-form representation is more suitable for such settings, as it clearly represents the order of moves and the information available to each player (as depicted in Fig. 2b)”
> A normal form game can represent a game with sequential decisions, and an extensive form game can represent a matrix game: they are roughly equivalent in representation power (there are oddities that come up with imperfect recall.) Both the simultaneous and sequential games in 2.2.5 can be represented as either a normal form game or extensive form game.
> An extensive form game -is- probably a more appropriate representation for a sequential game, but because a normal form representation (and pure/mixed strategy representation) grows exponentially with the number of information sets.

---

> > ### Author Response · Authors · 2025-05-23
> > **Response to reviewer Jb4T (Part 1)**
> >
> > We sincerely thank the reviewer for the careful reading of the revised manuscript and for the detailed, constructive feedback. Your suggestions have continually improved the quality of our paper. We have now uploaded a further revised version of the manuscript, in which all additional changes are highlighted in **purple**.
> >
> > > About the framework
> >
> > We appreciate the reviewer’s clarification and concerns. Our formalism is not intended to replace existing taxonomies familiar to self-play experts. Instead, it is meant as an entry point for readers who are not familiar with the field. Indeed, as noted in the discussion, we expect self-play methods to drive progress in much broader settings—including large-language-model training (§5.4) and real-world decision-making problems (§5.5). The proposed framework provides a single, concrete “handle” that helps those newcomers navigate the landscape. Once that handle is established, we use it to motivate the four categories presented in Section 3, even though some algorithms in those families (e.g., XDO) do not fit within the framework’s formal boundaries.
> >
> > To show that the framework can generate ideas rather than merely classify them, we have expanded §3.6.3 with another paragraph on cross-category inspiration. In particular, we highlight the recent cross-pollination between regret-minimization and PSRO research. Concretely, we illustrate the ODO [1] algorithm shows how population-based method can be augmented with regret-minimization, and we trace that lineage directly from the framework’s primitives. Such examples demonstrate the formalism’s capacity to surface fruitful research questions.
> >
> > [1] Dinh, L. C., et al. "Online double oracle." *Transactions on Machine Learning Research* (2022).
> >
> > > "further exacerbating the non-stationarity problem” claim needs justification? cooperative self-play also has problems with the strategies being non-stationary.
> >
> > We appreciate the reviewer’s concern and have clarified the statement accordingly: "In MARL, non-cooperative tasks are generally harder than cooperative ones because the absence of a shared reward prevents the effective use of centralized training, a key technique for mitigating non-stationarity in cooperative settings [1, 2, 3, 4], thereby further aggravating non-stationarity."
> >
> > [1] Foerster, Jakob, et al. "Learning to communicate with deep multi-agent reinforcement learning." *Advances in neural information processing systems* 29 (2016).
> >
> > [2] Foerster, Jakob, et al. "Counterfactual multi-agent policy gradients." *Proceedings of the AAAI conference on artificial intelligence*. Vol. 32. No. 1. 2018.
> >
> > [3] Lowe, Ryan, et al. "Multi-agent actor-critic for mixed cooperative-competitive environments." *Advances in neural information processing systems* 30 (2017).
> >
> > [4] Yu, Chao, et al. "The surprising effectiveness of ppo in cooperative multi-agent games." *Advances in neural information processing systems* 35 (2022): 24611-24624.
> >
> > > “focuses on its original and most prevalent application” Is it really the most prevalent? Applications to team coordination: drones, warehouse pathing, and distributed planning seem common, and could arguably be more commonly applied.
> >
> > Thank you for pointing this out. We agree that our original wording overstated the case. We removes the phrase “most prevalent” and revises the sentence in the introduction to read: "our survey is conducted within non-cooperative settings. "
> >
> > > “with a particular focus on non-cooperative settings” “within non-cooperative settings” it’s a restriction, not just a focus
> >
> > The word “focus” is imprecise here, because our study restricts itself to non-cooperative games rather than merely highlighting them. We have therefore replaced this wording wherever it occurs.

---

> > ### Author Response · Authors · 2025-05-23
> > **Response to reviewer Jb4T (Part 2)**
> >
> > > "The normal-form representation is appropriate here, as it captures the strategic interaction under imperfect information.” “Games of this nature are called dynamic games. The extensive-form representation is more suitable for such settings, as it clearly represents the order of moves and the information available to each player (as depicted in Fig. 2b)” A normal form game can represent a game with sequential decisions, and an extensive form game can represent a matrix game: they are roughly equivalent in representation power (there are oddities that come up with imperfect recall.) Both the simultaneous and sequential games in 2.2.5 can be represented as either a normal form game or extensive form game. An extensive form game -is- probably a more appropriate representation for a sequential game, but because a normal form representation (and pure/mixed strategy representation) grows exponentially with the number of information sets.
> >
> > We appreciate the reviewer’s detailed clarification. Our original wording risked implying that normal-form and extensive-form games differ in expressive power, when in fact any finite game can be converted from one form to the other. Our intent was only to highlight convenience and scalability, not capability. To remove this ambiguity, we have revised §2.2.5 to state explicitly that every game can be represented in either normal or extensive form, and we now explain why the extensive form is often preferred for dynamic games while the normal form remains more compact for static games.

---

### Author Response · Authors · 2025-05-10
**General response**

We would like to sincerely thank all the reviewers for taking the time to read and evaluate our manuscript. We greatly appreciate your constructive comments and insightful suggestions, which have significantly helped us improve the quality and clarity of the paper. We have uploaded a revised version of the manuscript. **All changes made in the revised manuscript are highlighted in blue.** In addition, we have provided a detailed point-by-point response addressing each of the reviewers’ comments. We welcome any further discussion or clarification.

---

### Decision · Action_Editor_w2Nw · 2025-05-29

**Recommendation:** Reject

**Comment:**

In the official recommendations, one reviewer states:

------------------------------------------------------------------------

"Initially, I was uncertain whether MARL and game-theoretic algorithms could be accurately classified under the umbrella of self-play. Later, the definition of self-play was broadened to be more inclusive. However, it still appears that existing algorithms are grouped under self-play (as well as their self-player framework) in a somewhat forceful manner. The framework itself seems to overlap significantly with PSRO—perhaps by as much as 80%—which is not surprising given that PSRO is already a highly general framework.

Second, I remain skeptical that their self-play framework offers meaningful insights for developing new algorithms. The insights they mention in the comments do not clearly stem from the framework itself but rather seem to emerge naturally from a solid understanding of the existing algorithms."

------------------------------------------------------------------------

Another reviewer states:

------------------------------------------------------------------------

"My opinion is largely summed up by "I’m not sure I understand what the framework is providing within the survey"

It seems to make some the algorithms less understandable: different both than what I would consider a natural description, and different than how they're originally described. The authors don't do anything with the proposed framework. I don't think they can do anything: having what is basically a "stick anything you want in there" component makes it rather non-specific. The framework takes up space and effort, and some of the algorithms end up under-described, in my opinion.

I also think there are some issues of correctness and completeness (or overly ambitious scope)."

------------------------------------------------------------------------


I agree with these points.

First: PSRO is not self-play. Self-play has a common understanding from 20-30 years of use, referring to Tesauro-style TD Gammon or AlphaZero: every agent uses the same algorithm and they all play against themselves. Game-theoretic methods like PSRO are much more complex, involving populations of policies, meta-analysis across their combinations, different sampling strategies to combine different opponents, etc. PSRO is closer to population-based training / population play, which is distinctly different from self-play.

Second, looking at Algorithm 3.1, it is basically the same as PSRO. The survey paper by Bighashdel et al. already summarizes all of the work on PSRO. As mentioned in reviews and official recommendations, it is not clear what benefit there is to stretching the definition to be so loose that it includes PSRO.

The motivation for having such a broad framework is unclear: the motivations behind game-theoretic RL algorithm,s such as NFSP, PSRO, etc. was specifically to draw inspiration from game-theoretic foundations, since traditional self-play as insufficient to address the pitfalls. I strongly recommend the authors improve their motivation for making a framework so broad (and perhaps also call it something different from "self-play" which has a decades-old well-accepted meaning in the community.)

**Audience:**

Audience is appropriate for TMLR

**Claims And Evidence:**

Since this is a survey paper, there are no claims in the traditional sense.

There is one statement: "While these studies are valuable, they do not provide a comprehensive perspective that fully captures the breadth and depth of self-play"

Two of three reviewers, and myself, disagree on the usefulness that such a comprehensive perspective provides (details below). In that sense, the statement is not adequately supported by the paper content.